# LLaCA: Multimodal Large Language Continual Assistant

## Abstract

Instruction tuning guides the Multimodal Large Language Models (MLLMs) in aligning different modalities by designing text instructions, which seems to be an essential technique to enhance the capabilities and controllability of foundation models. In this framework, Multimodal Continual Instruction Tuning (MCIT) is adopted to continually instruct MLLMs to follow human intent in sequential datasets. We observe existing gradient update would heavily destroy the tuning performance on previous datasets and the zero-shot ability during continual instruction tuning. Exponential Moving Average (EMA) update policy owns the ability to trace previous parameters, which can aid in decreasing forgetting. However, its stable balance weight cannot deal with the ever-changing datasets, leading to the out-of-balance between plasticity and stability of MLLMs. In this paper, we propose a method called Multimodal **L**arge **La**nguage **C**ontinual **A**ssistant (LLaCA) to address the challenge. Starting from the trade-off prerequisite and EMA update, we propose the plasticity and stability ideal condition. Based on Taylor expansion in the loss function, we find the optimal balance weight is basically according to the gradient information and previous parameters. We automatically determine the balance weight and significantly improve the performance. Through comprehensive experiments on LLaVA-1.5 in a continual visual-question-answering benchmark, compared with baseline, our approach not only highly improves anti-forgetting ability (with reducing forgetting from 22.67 to 2.68), but also significantly promotes continual tuning performance (with increasing average accuracy from 41.31 to 61.89). Our code will be published soon.

## 1 Introduction

Multimodal Large Language Modals (MLLMs) have demonstrated remarkable capabilities in vision-language understanding and generation (Li et al., 2023; Zhu et al., 2023; Achiam et al., 2023). It generally exits two stages: large scale pre-training and instruction-tuning. Instruction tuning is extremely significant due to guiding MLLMs in following human intent and aligning different modalities (Liu et al., 2024b; Chen et al., 2024b; Zheng et al., 2023), which seems to be an essential technique to enhance the capabilities and controllability of foundation models. In general, it enables a unified fine-tuning of image-instruction-output data format and makes the MLLMs easier to generalize to unseen data (Li et al., 2024; Liu et al., 2023).

As knowledge continuously evolves with the development of human society, new instructions are constantly generated, *e.g.* the emergence of new concepts or disciplines. How to enable existing MLLMs to assimilate these novel instructions and undergo self-evolution becomes the key challenge (Zhu et al., 2024). To accommodate the new instructions, the most effective strategy is incorporating both old and new instructions for joint training. However, even such relatively lightweight fine-tuning is unaffordable. Most importantly, directly fine-tuning these new instructions would destroy the pre-training knowledge, *e.g.* catastrophic forgetting (Goodfellow et al., 2013; Li et al., 2019; Nguyen et al., 2019).

Multimodal continual instruction tuning (MCIT) is proposed to address this challenge (He et al., 2023; Chen et al., 2024a). For MLLMs, previous methods, EProj (He et al., 2023), FwT(Zheng et al., 2024), and CoIN (Chen et al., 2024a), utilize the model expansion framework by continually adding new branches for the novel instructions, which therefore has less impact on the old knowledge.

However, they suffer from memory explosion and high computational cost problems. On the other hand, continually full fine-tuning (FFT) downstream datasets with single branch architecture would destroy the pre-trained parameters, and greatly reduce the zero-shot generalization performance of MLLMs (Zhai et al., 2023).

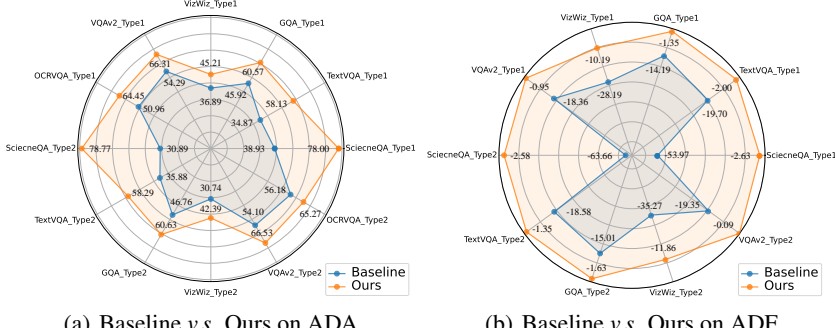

(a) Baseline *v.s.* Ours on ADA      (b) Baseline *v.s.* Ours on ADF

Figure 1: Radar chart of comparisons in terms of (a) Average Dataset Accuracy (simplified as ADA, higher is better) and (b) Average Dataset Forgetting (simplified as ADF, lower is better) between baseline and ours. Detailed information about ADA and ADF please refer to Appendix A.9. Type1 and Type2 represent two distinct instructions, which will be specifically illustrated in section 5.

Considering the essential mechanism of parameter update, we discover that the gradient update might not be a satisfactory choice for MCIT. First of all, we find that the gradient inevitably drives the update of parameters toward the optimization of the new dataset, which causes forgetting. Instead, EMA adopts a weighted summation between old and new model parameters, which enjoys the natural advantage of keeping old knowledge. However, it faces the challenge of balancing old and new knowledge in continual tuning, as a fixed EMA weight cannot adapt to the continuously evolving datasets, *e.g.* from location identification dataset GQA to OCR token recognition dataset OCRVQA. To determine this balance weight, it seems that the gradient actually represents the magnitude or discrepancy between the model and the new instructions. Thus we propose a new ideal condition for MCIT. We use Taylor expansion in the loss function by assuming our update holds the ideal condition, and find the optimal weight is basically according to the gradient information.

In this paper, we propose a novel method: Multimodal **L**arge **La**nguage **C**ontinual **A**ssistant (LLaCA). We introduce the stability and plasticity ideal state to formulate simultaneous equations with the EMA update. By employing the Lagrange multiplier method to solve the equation set, we provide a theoretical derivation for dynamically updating the weight of EMA in each training iteration. In order to achieve a simple but highly effective update method, we further propose two approximate compensation mechanisms.

We adopt LLaVA-1.5 (Liu et al., 2024a) as the backbone and insert the efficient-tuning parameters: LoRA (Hu et al., 2022) in the LLM (7B-Vicuna). In the whole continual tuning process, only one set of LoRA parameters and a projection layer are trained by incorporating our method, which can greatly reduce the memory and computation burden, as shown in Figure 2(a). We continually fine-tune on visual-question-answering datasets and verify performances of the tuned model on both in- and out-of-domain datasets. Experimental results consistently demonstrate excellent anti-forgetting, continual tuning, and zero-shot generalization performance of our method. Additionally, without saving exemplars or expanding modules, our method can be free of privacy leakage or memory explosion. In summary, the contributions of this paper are as follows:

**Alleviating Catastrophic Forgetting in MLLMs.** Through rigorous deduction, we propose a novel method of reducing catastrophic forgetting and meanwhile keeping well new knowledge assimilating in multimodal continual instruction tuning.

**Generalized Application and Few Tuning Costs.** Our proposed method is model-free and can be easily applied in a wide range of fine-tuned methods and MLLMs. Additionally, it costs few tuning resources, due to introducing two approximations.

**State-of-The-Art Performance.** To the best of our knowledge, our method shows the best comprehensive continual tuning performance compared with others, especially significant enhancements in

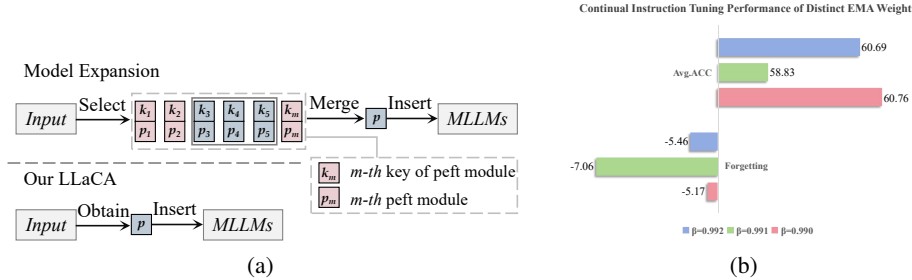

(a)                                                                                              (b)

Figure 2: (a) The PEFT module design of LLaCA *v.s.* Model Expansion method. Above: Model Expansion method, which queries and merges PEFT module from a pool of learnable PEFT modules, *e.g.* Prompt, LoRA, Projection Layer. Below: LLaCA, which only has one learnable PEFT module and obtains it directly. and (b) MCIT performance of stable EMA summation weight.

the performance of the backbone (as shown in Figure 1). Besides that, our method also owns both practicality and robustness.

## 2 RELATED WORK

**Multi-modal Large Language Models.** MLLMs own strong reasoning and complex contextual understanding capabilities, which can generate text sequences according to the input images and texts. With a frozen large language model and visual tower, BLIP2 (Li et al., 2023) bridged the gap between image and text modality through Qformer structure. Inspired by the notable reasoning power of instruction tuning in LLMs, *e.g.,* LLaMA (Touvron et al., 2023) and GPT (Floridi & Chiriatti, 2020; Achiam et al., 2023), LLaVA (Liu et al., 2024b) and MiniGPT4 (Zhu et al., 2023) adopted instruction tuning with only training a linear projection layer to align the image-text modalities. Recently, LLaVA-1.5 (Liu et al., 2024a) and MiniGPT5 (Zheng et al., 2023) refined the instructions and improved the performance across a wider range of multi-modality datasets.

**Multimodal Continual Instruction Tuning Work:** (Zhang et al., 2023) first proposed the continual instruction tuning with LLMs. Following it, (He et al., 2023) and (Chen et al., 2024a) designed distinct cross-modality continual instruction tuning benchmarks with InstructBLIP and LLaVA, respectively. Specifically, (He et al., 2023) adopted unique linear projection layers for each dataset and utilized a key-query driven mechanism to infer the dataset identifier. (Chen et al., 2024a) employed the MOELoRA paradigm, assigning different LoRA weights to different datasets based on expert knowledge. Nevertheless, they still faced the issue of forgetting due to repeated training.

## 3 PRELIMINARY

**Multimodal Continual Instruction Tuning Definition:** MCIT (He et al., 2023) is defined to leverage MLLMs to continually instruction-tune on new datasets without costly re-training. Compared with traditional continual learning, MCIT differs in that it pays more attention to effectively leveraging the rich natural language instructions to prevent catastrophic forgetting and encourage knowledge transfer. MCIT can be described as a set of datasets $\mathcal{T}_{\text{seq}} = \{t_1, ..., t_T\}$ that arrives sequentially. Note that datasets in the stream can be any type and are not restricted to specific categories or domains. Each dataset $t_j \in \mathcal{T}_{\text{seq}}$ has a natural language instruction $I^{t_j}$, training set $\mathcal{D}_{\text{train}}^{t_j}$ and test set $\mathcal{D}_{test}^{t_j}$. The goal of MCIT is to learn a single model $f$ from $\mathcal{T}_{\text{seq}}$ sequentially.

**Brief Review of The EMA Update Policy:** The EMA update has two kinds of parameters [1], one is parameters $\theta$ updating normally with gradient, another is EMA parameters $\theta^*$ updating as:

$$\theta_t^* = \beta_t \theta_{t-1}^* + (1 - \beta_t)\theta_t, \tag{1}$$

where $\beta_t$ is the EMA weight, $t$ and $t - 1$ is the training iteration. According to Eq.(1), performance on the current training iteration of $\theta_t^*$ is worse than $\theta_t$ because it only transfers a portion of the new

---

[1]Without a specific illustration, parameters in this paper refer to trainable parameters.

knowledge from $\theta_t$ to $\theta_t^*$. Meanwhile, by mathematical induction from Eq.(1), an equivalent and unified equation can be marked as:

$$\theta_t^* = (\prod_{i=1}^{t} \beta_i) \cdot \theta_0 + \sum_{i=1}^{t} [(1 - \beta_i) \cdot (\prod_{j=i+1}^{t} \beta_j) \cdot \theta_i]. \tag{2}$$

Detailed process please kindly refer to Appendix A.1. Eq.(2) implies that $\theta^*$ is a weighted sum of $\theta_i (i \in \{1, t\})$, and the summation weight is a product about $\beta_i$ in different iterations. Each update can contribute to the EMA parameters by reviewing the previous parameters, which can make it have excellent stability. While for the traditional gradient update, the gradient only carries novel information, but without reviewing the previous one.

From Figure 2(b), we further discover that the performance of the EMA method is greatly affected by the summation weight, and a stable EMA weight cannot be applied to flexible and various instructions, *e.g.* "Answer the question using a single word or phrase" in OCRVQA dataset and "Answer with the option's letter from the given choices directly" in ScienceQA dataset (Mishra et al., 2019; Hudson & Manning, 2019). As a consequence, we are motivated to propose a dynamical update method for EMA weight.

## 4 METHOD

### 4.1 PROPOSITION OF OUR METHOD

Primarily, we propose the following equations to simultaneously achieve the optimal ideal state with EMA update (Van de Ven & Tolias, 2019).

**Proposition 4.1** *(Ideal State). Given an MLLM with multimodal continual instruction tuning, with its parameters $\theta$ and EMA parameters $\theta^*$, after training on the iteration $t$, we can describe the best new knowledge transferring and the best old knowledge protecting as:*

$$\begin{cases} \mathcal{L}(\theta_t^*, x_t) = \mathcal{L}(\theta_t, x_t), \textit{(best new knowledge transferring)} \\ \quad\quad \theta_t^* = \theta_{t-1}^*. \textit{(best old knowledge protecting)} \end{cases} \tag{3}$$

The first equation of Eq.(3) represents ensuring the model performance on the new dataset with no change of training loss, inspired by The Optimal Brain Surgeon framework (Hassibi et al., 1993; LeCun et al., 1989; Frantar & Alistarh, 2022; Molchanov et al., 2022). The second equation of Eq.(3) represents preserving the model performance on old datasets with no change of model parameters.

***Discussion***: From Proposition.4.1, we can further find that the starting point is model-independent. Thus our method can be extended to more MLLMs, and parameter-efficient tuning paradigms, even more continual learning scenes.

### 4.2 SELF-ADAPTION DYNAMICAL UPDATE METHOD

In order to find a dynamical update $\beta_t$, and realize Proposition.4.1, we start from a Taylor expansion of the loss function $\mathcal{L}$ around the individual parameter $\theta_t$ [2]. Our basis is that the gradient, which represents the discrepancy between the parameters and the new knowledge, is generated by the loss function. To further introduce $\theta_t^*$ in the Taylor expansion, we replace $\theta$ with $\theta_t^*$, and have:

$$\mathcal{L}(\theta_t^*) - \mathcal{L}(\theta_t) = \mathcal{L}'(\theta_t)(\theta_t^* - \theta_t) + \frac{\mathcal{L}''(\theta_t)}{2}(\theta_t^* - \theta_t)^2. \tag{4}$$

Notice that we have omitted the high-order infinitesimal term of $O(\theta - \theta_t)^3$. Additionally, we introduce the relaxation factor $\Delta\theta$ and have the stability constraint that $\theta_t^* = \theta_{t-1}^* + \Delta\theta$. Here, our intuition is from the perspective of EMA parameters update. The relaxation factor $\Delta\theta$ denotes the newly assimilated model parameters. Moving the left item to the right of the equation, we have:

$$\Delta\theta + \theta_{t-1}^* - \theta_t^* = 0. \tag{5}$$

---

[2]For simplification, we omit the $x_t$ in $\mathcal{L}$

Start from the stability constraint, combined with Eq.(1), we can obtain:

$$\theta_t^* - \theta_t = -\frac{\beta_t}{1 - \beta_t}\Delta\theta = \frac{\beta_t}{\beta_t - 1}\Delta\theta. \tag{6}$$

Please kindly refer to Appendix A.2 for detailed demonstrations. In order to achieve the best new knowledge transferring and the best old knowledge protecting, we minimize the difference between $\mathcal{L}(\theta_t^*)$ and $\mathcal{L}(\theta_t)$, $\theta_t^*$ and $\theta_{t-1}^*$. Merging the two minimal situations, we have a unified optimal objective function and set up the following minimization problem:

$$\min \{\mathcal{L}(\theta_t^*) - \mathcal{L}(\theta_t) + \theta_t^* - \theta_{t-1}^*\}, \\ s.t. \Delta\theta + \theta_{t-1}^* - \theta_t^* = 0. \tag{7}$$

To further consider the constrained minimization problem, we use the method of Lagrange multipliers, which combines the objective function with the constraint by incorporating the Lagrange multiplier $\lambda$.

$$F = \mathcal{L}(\theta_t^*) - \mathcal{L}(\theta_t) + \theta_t^* - \theta_{t-1}^* + \lambda(\Delta\theta + \theta_{t-1}^* - \theta_t^*). \tag{8}$$

From Eq.(1), we can transfer the *s.t.* equation as:

$$\Delta\theta + \theta_{t-1}^* - \theta_t^* = \Delta\theta + \theta_{t-1}^* - [\beta_t\theta_{t-1}^* + (1 - \beta_t)\theta_t] = \Delta\theta + (1 - \beta_t)(\theta_{t-1}^* - \theta_t). \tag{9}$$

After that, we substitute Eq.(4), Eq.(5), Eq.(6) and Eq.(9) into Eq.(8).

$$F = \mathcal{L}'(\theta_t)\frac{\beta_t}{\beta_t - 1}\Delta\theta + \frac{\mathcal{L}''(\theta_t)}{2}(\frac{\beta_t}{\beta_t - 1}\Delta\theta)^2 + \Delta\theta + \lambda[\Delta\theta + (1 - \beta_t)(\theta_{t-1}^* - \theta_t)]. \tag{10}$$

Taking the derivative of the Lagrangian concerning $\beta_t$, we set it to zero and determine the direction in which the Lagrangian is stationary. This condition is essential for finding the optimal solution.

$$\frac{\partial F}{\partial \beta_t} = -\frac{1}{(\beta_t - 1)^2}\mathcal{L}'(\theta_t)\Delta\theta - \frac{\beta_t}{(\beta_t - 1)^3}\mathcal{L}''(\theta_t)\Delta\theta^2 - \lambda(\theta_{t-1}^* - \theta_t) = 0. \tag{11}$$

By solving these equations, we obtain one feasible solution for $\beta_t$, which minimizes the objective function while satisfying the constraint:

$$\beta_t = \frac{\mathcal{L}'(\theta_t) + 1}{(\theta_t - \theta_{t-1}^*)\mathcal{L}''(\theta_t)}. \tag{12}$$

Please kindly refer to Appendix A.3 for the detailed deduction.

***Discussion***: Based on Eq.(12), we can discover that the optimal weight is meanwhile basically related to the new gradient $\mathcal{L}'$ and the old EMA parameters $\theta_{t-1}^*$. It illustrates that the obtained EMA weight $\beta_t$ can make the trade-off between learning new knowledge and alleviating the forgetting of old knowledge.

### 4.3 TWO APPROXIMATE OPTIMIZATIONS

In Eq.(12), the calculation of $\mathcal{L}''(\theta_t)$ involves the inverse of the Hessian matrix, which needs to obtain second-order partial derivatives. However, the above calculation is complex, leading to expensive memory and time-consuming, let alone for LLM, which further increases the training burden. Thus, how to approximately express the Hessian matrix without a complex calculation process becomes a challenge and urgently needs to be solved.

**Approximate Optimization Step I:** Considering that the Hessian matrix is obtained by partially deriving the gradients, we can approximate the derivative with the quotient of the differentiation. Here we recognize each iteration as a period of parameter update and further simplify the denominator. As a result, we have the following approximation equation to estimate the $\mathcal{L}''(\theta_t)$.

$$\mathcal{L}''(\theta_t) = \frac{\partial \mathcal{L}'(\theta_t)}{\partial \theta_t} = \frac{\mathcal{L}'(\theta_t) - \mathcal{L}'(\theta_{t-1})}{\theta_t - \theta_{t-1}}, \tag{13}$$

where the $\mathcal{L}'(\theta_t)$ represents gradients in the current iteration, and the $\mathcal{L}'(\theta_{t-1})$ denotes gradients in the last iteration.

$\theta_t$ in Eq.(12) represents the individual parameter, which causes the EMA weight $\beta_t$ to be individual parameter-wise. However, the training parameters $\theta$ always own a high dimension, leading to a huge computational cost. Thus, we propose to set the union parameter-wise $\beta_t$, *e.g.* one $\beta_t$ for one module layer, and utilize the following method to approximate the $\beta_t$.

**Approximate Optimization Step II:** Assuming that the EMA weight of each individual parameter in the same module layer would not change a lot. Here, we introduce L1-Norm and further approximate the $\beta_t$ as:

$$\beta_t \approx \|\frac{[\mathcal{L}'(\hat{\theta}_t) + 1](\hat{\theta}_t - \hat{\theta}_{t-1})}{(\hat{\theta}_t - \hat{\theta}_{t-1}^*)[\mathcal{L}'(\hat{\theta}_t) - \mathcal{L}'(\hat{\theta}_{t-1})]}\|. \tag{14}$$

$\hat{\theta}$ represents the whole module layer parameters. If we have a coarse approximation that $\hat{\theta}_{t-1} = \hat{\theta}_{t-1}^*$, we can discover that:

$$\beta_t \approx \|\frac{\mathcal{L}'(\hat{\theta}_t) + 1}{\mathcal{L}'(\hat{\theta}_t) - \mathcal{L}'(\hat{\theta}_{t-1})}\| > 1. \tag{15}$$

Considering that the EMA weight $\beta_t \in (0, 1)$, thus we adopt the $\beta_t = 0.99$ when the $\beta_t$ exceeds the range of $(0, 1)$, where the constant value 0.99 is empirically obtained from experiments.

***Discussion*:** The motivations of adopting L1-Norm to approximate the $\beta_t$ are: 1) L1-Norm owns the extraordinary performance of approximating the $\beta_t$. 2) L1-Norm occupies a few computation loads. For the detailed implementation of our method please refer to Appendix A.7.

## 4.4 OVERVIEW OF OUR METHOD

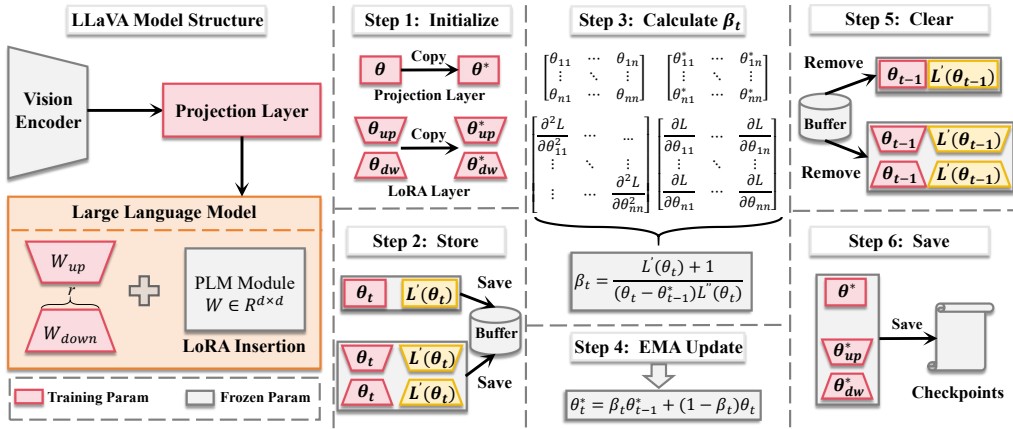

Figure 3: Overview of the LLaCA method. It is mainly divided into six steps. Step 1: Initialize the EMA parameters. Step 2: Store the gradients and parameters. Step 3: Calculate the EMA weight. Step 4: Update the EMA parameters. Step 5: Clear the former storage. Step 6: Save the EMA parameters.

Our method consists of six primary steps, which are shown in Figure 3. Before starting training, the initial step is creating EMA parameters $\theta^*$ and copying parameters $\theta$ to $\theta^*$. After the training of iteration $t$, gradients $\mathcal{L}'(\theta_t)$ and model parameters $\theta_t$ would be saved and involved in the EMA weight calculation process of the next iteration $t + 1$. Based on Eq.(14), we can obtain the adaption EMA weight $\beta_t$. With the EMA weight $\beta_t$, we can update the EMA parameters from $\theta_{t-1}^*$ to $\theta_t^*$ based on Eq.(1). After the training of iteration $t$, to reduce the memory burden, we will clear the saved gradients $\mathcal{L}'(\theta_{t-1})$ and model parameters $\theta_{t-1}$. After training in each downstream dataset, we only save the EMA parameters $\theta^*$ and use them to test the performance in the previous and current datasets. Additionally, we also adopt them to initialize the parameters in the next tuning dataset. For the corresponding detailed algorithm process please kindly refer to Appendix A.12.

Table 1: Avg.ACC, Forgetting, and New.ACC performance, obtained with instruction of type 1.

| Method | Venue | Datasets | | | | | | Metrics | | |
|---|---|---|---|---|---|---|---|---|---|---|
| | | ScienceQA | TextVQA | GQA | VizWiz | VQAv2 | OCRVQA | Avg.ACC(↑) | Forgetting(↓) | New.ACC(↑) |
| Zero-shot | - | 49.91 | 2.88 | 2.08 | 0.90 | 0.68 | 0.17 | 9.44 | - | - |
| Fine-Tune | - | 25.72 | 30.56 | 38.49 | 34.42 | 44.75 | 58.84 | 38.80 | 26.87 | 61.62 |
| LoRA(Baseline)(Hu et al., 2022) | ICLR'22 | 51.50 | 32.38 | 38.62 | 29.27 | 45.11 | 50.96 | 41.31 | 22.67 | 60.20 |
| MoELoRA(Chen et al., 2024a) | ArXiv'24 | 54.09 | 45.21 | 37.65 | 37.58 | 43.48 | 59.74 | 44.61 | 21.03 | 62.13 |
| LWF(Li & Hoiem, 2017) | TPAMI'16 | 61.51 | 34.68 | 37.23 | 32.97 | 39.10 | 50.82 | 42.72 | 15.17 | 55.52 |
| EWC(Kirkpatrick et al., 2017) | PNAS'17 | 68.19 | 36.71 | 37.18 | 33.53 | 40.73 | 50.61 | 44.49 | 12.69 | 55.07 |
| PGP(Qiao et al., 2024a) | ICLR'24 | 73.31 | 51.41 | 57.34 | 47.46 | 60.69 | 56.91 | 57.85 | 3.22 | 60.54 |
| **LLaCA(Ours)** | - | **77.15** | **56.54** | **60.18** | **47.16** | **65.83** | **64.45** | **61.89** | **2.68** | **64.12** |
| Upper-Bond | - | 80.19 | 59.69 | 61.58 | 52.00 | 66.78 | 64.45 | 64.12 | - | - |

Table 2: Avg.ACC, Forgetting, and New.ACC performance, obtained with instruction of type 2.

| Method | Venue | Datasets | | | | | | Metrics | | |
|---|---|---|---|---|---|---|---|---|---|---|
| | | ScienceQA | TextVQA | GQA | VizWiz | VQAv2 | OCRVQA | Avg.ACC(↑) | Forgetting(↓) | New.ACC(↑) |
| Zero-shot | - | 49.91 | 3.31 | 3.02 | 0.85 | 0.68 | 1.05 | 9.80 | - | - |
| Fine-Tune | - | 28.36 | 27.43 | 34.18 | 27.66 | 41.03 | 54.38 | 35.51 | 26.92 | 60.06 |
| LoRA(Baseline)(Hu et al., 2022) | ICLR'22 | 39.00 | 32.34 | 38.02 | 15.33 | 44.42 | 56.18 | 37.55 | 28.32 | 61.15 |
| MoELoRA(Chen et al., 2024a) | ArXiv'24 | 39.42 | 35.26 | 37.17 | 17.94 | 44.57 | 57.34 | 38.62 | 28.31 | 62.21 |
| LWF(Li & Hoiem, 2017) | TPAMI'16 | 21.14 | 32.16 | 35.08 | 10.27 | 36.43 | 57.09 | 32.03 | 28.29 | 58.34 |
| EWC(Kirkpatrick et al., 2017) | PNAS'17 | 20.30 | 32.75 | 34.92 | 10.51 | 38.20 | 56.86 | 32.26 | 28.11 | 55.68 |
| PGP(Qiao et al., 2024a) | ICLR'24 | 72.88 | 48.99 | 56.49 | 43.44 | 60.12 | 55.56 | 56.25 | 4.40 | 59.92 |
| **LLaCA(Ours)** | - | **77.91** | **57.15** | **60.38** | **40.19** | **66.48** | **65.27** | **61.23** | **3.38** | **64.05** |
| Upper-Bond | - | 80.92 | 59.37 | 61.85 | 50.29 | 66.57 | 65.27 | 64.05 | - | - |

# 5 EXPERIMENTS

## 5.1 EXPERIMENTAL SETUP

**Implementation:** We adopt LLaVA-1.5 (Liu et al., 2024b) as our backbone with inserted LoRA (Hu et al., 2022) in the LLM (Vicuna-7B). During the whole multimodal continual instruction tuning, we freeze the vision encoder and LLM, with only training the projector and LoRA. We follow the datasets of the CoIN benchmark (Chen et al., 2024a), including ScienceQA (Lu et al., 2022), TextVQA (Singh et al., 2019), GQA (Hudson & Manning, 2019), VizWiz (Gurari et al., 2018), VQAv2 (Goyal et al., 2017) and OCR-VQA (Mishra et al., 2019). We keep the training order that: ScienceQA, TextVQA, GQA, VizWiz, VQAv2, and OCR-VQA. Additionally, we validate all the methods on two distinct instructions. The first is normal and only owns one kind of instruction template. While the other would be more challenging due to being equipped with ten kinds of instruction templates. All experiments are conducted on 8 NVIDIA A100 GPUs, excluding Table 3.

**Compared Methods:** We compare the LLaCA method against eight methods including (1) zero-shot and (2) full fine-tune performance from (Chen et al., 2024a); (3) LoRA (Hu et al., 2022) as our baseline (throughout the paper, the baseline consistently refers to LoRA Fine-Tuning), which prepends LoRA parameter efficient tuning paradigm into LLM and only tunes the linear projector and LoRA parameters; (4) MoELoRA (Chen et al., 2024a); (5) LWF (Li & Hoiem, 2017); (6) EWC (Kirkpatrick et al., 2017); (7) PGP (Qiao et al., 2024a); (8) Upper-bond, the best accuracy in each dataset. Notice that all the methods, except zero-shot and fine-tuning, are based on the LoRA method. The detailed descriptions of the above methods can be found in Appendix A.6.

**Evaluation Metrics:** We follow the most popular protocols for evaluation (Wang et al., 2022b;a; Smith et al., 2023; Qiao et al., 2024b), which are Average Accuracy (Simplified as Avg.Acc), Forgetting, and New Accuracy (Simplified as New.Acc) (please refer to Appendix A.9 for more details).

## 5.2 MULTIMODAL INSTRUCTION TUNING RESULTS

**Comparison to SOTA:** We compare the performance of our method with others in Table 1. We observe that LLaCA can greatly improve the Avg.ACC (+20.58) and reduce the Forgetting (-19.99) compared with the baseline, demonstrating its excellent anti-forgetting and continual tuning ability. It is also noteworthy that LLaCA outperforms the best of other methods by +4.04@Avg.ACC, -0.54@Forgetting, and +3.58@New.ACC. Although methods like LWF (Li & Hoiem, 2017) and EWC (Kirkpatrick et al., 2017) can resist forgetting, their plasticity is greatly influenced. PGP (Qiao et al., 2024a) can perform well both in stability and plasticity, while the Avg.ACC still needs to be improved. It is highlighted that LLaCA owns the highest Avg.ACC, New.ACC and the lowest forgetting among these methods, which shows that LLaCA can achieve the best trade-off between plasticity and stability. Additionally, we also validate all the methods in the second instruction, as shown in Table 2, and our method also owns the best performance. Although validated on two

different types of instructions, the Avg.ACC is almost unchanged, which shows that our method owns well generalization ability and is suitable for many instructions.

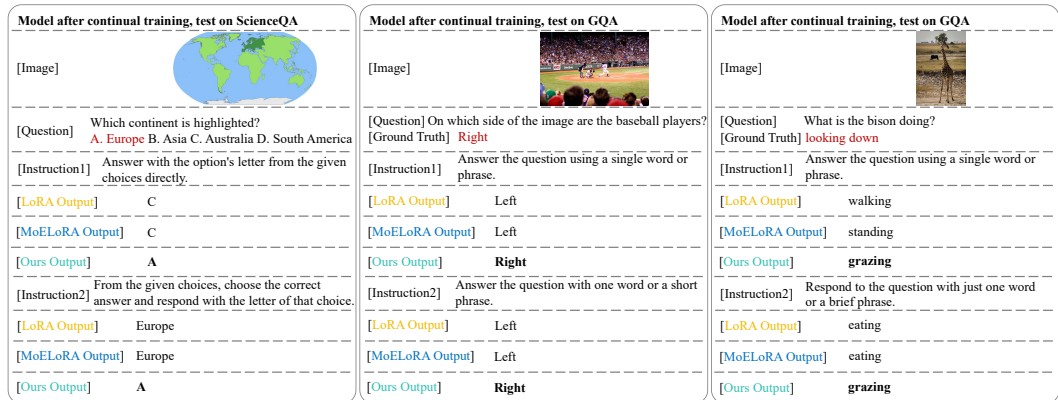

Figure 4: Visualization of multimodal continual instruction tuning examples, comparison between LoRA, MoELoRA, and ours.

**Analysis of Examples:** In Figure 4, we can observe that LLaCA can revise errors committed by baseline after continual tuning. For example, LLaCA can not only keep the geographic location recognition capability of MLLMs (as shown in the top left part) but also protect the knowledge related to the identification of direction and position in real scenarios (as shown in the middle part). Besides that, LLaCA can also be enlightening for MLLMs, as it can guide the model to generate more specific answers, such as changing the action of the bison in the top right image from a coarse description of "looking down" to a more detailed, informative "grazing", which presents that our method could expand the cognitive ability of MLLMs (as shown in the right part). Additionally, we are surprised to find that LLaCA can spontaneously suppress the occurrence of hallucinations appearing in the continual instruction tuning. (Zhai et al., 2023) deem that the hallucination in large models is related to forgetting in continual tuning. The above view is consistent with our experimental results. *e.g.* phenomenon that the answer of MLLMs without following the instruction, could be eliminated with reducing forgetting (as shown in the bottom left part). We also present the cases of multiple rounds of dialogue in Appendix A.4.

**Analysis of Training Time and Memory:** In Table 3, we compare the training memory and training time. Compared with LoRA, the baseline, training parameters of our method are kept the same, and the training time only increases 5.8%. However, these costs are worth it, as the Avg.ACC has improved by about 20% and the Forgetting has reduced by about 90%. Additionally, compared to the MoELoRA, our method not only owns the absolute advantage in performance but also holds fewer costs in training time, and training parameters.

Table 3: Comparison of training time, and training parameters. Experiments are conducted on 8 NVIDIA A6000 GPUs.

| Method | Training Time | | | | | | Training Params |
|--------|-----------|---------|---------|---------|---------|---------|-----------------|
| | ScienceQA | TextVQA | GQA | VizWiz | VQAv2 | OCRVQA | |
| LoRA | 13min | 48min | 3h10min | 24min | 2h30min | 3h47min | 1 x |
| MoELoRA | 20min | 1h | 4h | 32min | 3h10min | 4h43min | 3 x |
| **Ours** | 15min | 50min | 3h20min | 25min | 2h40min | 4h | 1 x |

Table 4: Zero-shot average accuracy performance after each dataset. Results are obtained with two instruction types, red marks unseen datasets (in-domain), and blue marks unseen datasets (out-of-domain) .

| Datasets of Instruction Type 1 | | | | | | Baseline | Ours |
|---|---|---|---|---|---|---|---|
| TextVQA | GQA | VizWiz | VQAv2 | OCRVQA | ImageNet | 6.68 | 47.55 |
| | GQA | VizWiz | VQAv2 | OCRVQA | ImageNet | 5.38 | 45.45 |
| | | VizWiz | VQAv2 | OCRVQA | ImageNet | 25.32 | 45.24 |
| | | | VQAv2 | OCRVQA | ImageNet | 17.89 | 43.62 |
| | | | | OCRVQA | ImageNet | 17.03 | 28.58 |
| | | | | | ImageNet | 3.75 | 7.82 |

| Datasets of Instruction Type 2 | | | | | | Baseline | Ours |
|---|---|---|---|---|---|---|---|
| TextVQA | GQA | VizWiz | VQAv2 | OCRVQA | ImageNet | 8.25 | 42.68 |
| | GQA | VizWiz | VQAv2 | OCRVQA | ImageNet | 5.20 | 44.65 |
| | | VizWiz | VQAv2 | OCRVQA | ImageNet | 23.53 | 44.56 |
| | | | VQAv2 | OCRVQA | ImageNet | 24.46 | 43.82 |
| | | | | OCRVQA | ImageNet | 19.16 | 30.18 |
| | | | | | ImageNet | 4.34 | 7.43 |

## 5.3 ZERO-SHOT PERFORMANCE

Catastrophic forgetting can severely impact the zero-shot generalization ability of large-scale pre-trained models, leading to the zero-shot collapse phenomenon (Zhai et al., 2023). Based on the

proposed LLaCA, we design experiments to validate its ability to mitigate zero-shot collapse in pre-trained MLLMs. In the experiment, we continually fine-tune on the six VQA datasets. After each fine-tuning, we would test the zero-shot inference ability on the unseen VQA (in-domain) datasets and ImageNet (out-of-domain) dataset (Deng et al., 2009). We adopt the average accuracy on the test datasets, and the higher score denotes the better zero-shot performance. The experimental results, as presented in Table 4, demonstrate that LLaCA can significantly enhance the zero-shot generalization ability of the MLLMs after continuous fine-tuning in downstream datasets compared to the baseline.

## 5.4 ROBUST PERFORMANCE

To further validate the robustness of the proposed method, we adopt four types of training strategies with various training orders and instructions (For detailed information please refer to Appendix A.10). Results are shown in the Figure 5(b). We can find that although New.ACC and Forgetting have a small range of fluctuations, but for Avg.ACC, as a comprehensive performance metric, its range of variation is almost invisible. We further research the average accuracy on each dataset in various types of training strategies and the results are shown in Figure 5(a). We can see that the average accuracy of the same dataset in each type looks very close to each other, which also proves the robustness of our method. Based on the above observations, we conclude that the fluctuations in New.ACC and Forgetting are caused by changes in instruction type and specific dataset training order. However, our method has strong robustness, which can maintain the Avg.ACC at a stable level in each type of training strategy.

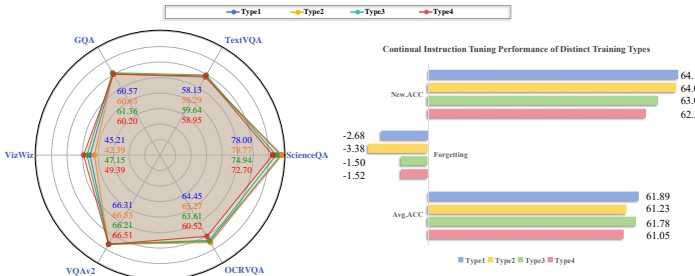

Figure 5: (a). Rader chart of Average Dataset Accuracy and (b). Histogram of Avg.ACC, Forgetting, and New.ACC performances of each training strategy type.

## 5.5 ABLATION STUDY

To validate the efficiency of the proposed method, we compare it with the baseline and traditional EMA method with a stable weight of 0.99. Results are shown in Table 5. Compared with the other two methods, our method owns the lowest forgetting and acquires the best comprehensive performance in Avg.ACC.

Table 5: Ablation study: comparison between baseline, stable EMA and ours.

| Instruction | Method | Avg.ACC(↑) | Forgetting(↓) | New.ACC(↑) |
|---|---|---|---|---|
| Type1 | Baseline | 41.31 | 22.67 | 60.20 |
| | Stable EMA | 60.76 | 5.17 | **65.08** |
| | **Ours** | **61.89** | **2.68** | 64.12 |
| Type2 | Baseline | 37.55 | 28.32 | 61.15 |
| | Stable EMA | 58.85 | 7.87 | **65.41** |
| | **Ours** | **61.23** | **3.38** | 64.05 |

## 6 CONCLUSION

To enable MLLMs to possess the ability of multimodal continual instruction tuning and further resist forgetting, we propose a novel method called LLaCA. Combined with the exponential moving average, the proposed method can protect previous knowledge and incorporate new knowledge at the same time. By solving a set of equations based on the Lagrange multiplier method, we obtain the self-adaption weight of EMA in each update process. Subsequently, two compensation mechanisms are further introduced to alleviate the computational costs. Experiments show that our approach not only owns excellent anti-forgetting and continual tuning ability but also well zero-shot performance. Additionally, our method needs a few extra computation costs and memory usage. Due to computational resource constraints, our current focus is primarily on continual instruction tuning. In the future, we aim to extend our method to continual pre-training scenarios.

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

## A APPENDIX

### A.1 DECOMPOSE OF EMA UPDATE

In the EMA update, it exists two kinds of parameters, normally parameters $\theta$ and EMA parameters $\theta^*$. At iteration 1, $\theta_1^*$ is updated according to Eq.(1) as:

$$\theta_1^* = \beta_1 \theta_0^* + (1 - \beta_1)\theta_1. \tag{16}$$

Then at iteration 2, by replacing Eq.(16), $\theta_2^*$ is updated as:

$$\begin{aligned}
\theta_2^* = \beta_2 \theta_1^* + (1 - \beta_2)\theta_2 &= \beta_2[\beta_1 \theta_0^* + (1 - \beta_1)\theta_1] + (1 - \beta_2)\theta_2 \\
&= \beta_2 \beta_1 \theta_0^* + \beta_2(1 - \beta_1)\theta_1 + (1 - \beta_2)\theta_2.
\end{aligned} \tag{17}$$

After that, at iteration 3, by replacing Eq.(17), $\theta_3^*$ is updated as:

$$\begin{aligned}
\theta_3^* = \beta_3 \theta_2^* + (1 - \beta_3)\theta_3 &= \beta_3[\beta_2 \beta_1 \theta_0^* + \beta_2(1 - \beta_1)\theta_1 + (1 - \beta_2)\theta_2] + (1 - \beta_3)\theta_3 \\
&= \beta_3 \beta_2 \beta_1 \theta_0^* + \beta_3 \beta_2(1 - \beta_1)\theta_1 + \beta_3(1 - \beta_2)\theta_2 + (1 - \beta_3)\theta_3.
\end{aligned} \tag{18}$$

Observing the equation form, based on the method of summarization and induction, we have the following assumption for iteration $n - 1$:

$$\theta_{n-1}^* = \prod_{i=1}^{n-1} \beta_i \cdot \theta_0^* + \sum_{i=1}^{n-1}(1 - \beta_i) \cdot \prod_{j=i+1}^{n-1} \beta_j \cdot \theta_i. \tag{19}$$

Finally, at iteration $n$, by replacing Eq.(19), $\theta_n^*$ is updated as:

$$\begin{aligned}
\theta_n^* &= \beta_n[\prod_{i=1}^{n-1} \beta_i \cdot \theta_0^* + \sum_{i=1}^{n-1}(1 - \beta_i) \cdot \prod_{j=i+1}^{n-1} \beta_j \cdot \theta_i] + (1 - \beta_n)\theta_n \\
&= \prod_{i=1}^{n} \beta_i \cdot \theta_0^* + \sum_{i=1}^{n-1}(1 - \beta_i) \cdot \prod_{j=i+1}^{n} \beta_j \cdot \theta_i + (1 - \beta_n)\theta_n \\
&= \prod_{i=1}^{n} \beta_i \cdot \theta_0^* + \sum_{i=1}^{n}(1 - \beta_i) \cdot \prod_{j=i+1}^{n} \beta_j \cdot \theta_i.
\end{aligned} \tag{20}$$

It can be found that Eq.(20) also has the same form as Eq.(19), which means that the assumption is established. Due to utilizing $\theta_0$ to initialize $\theta_0^*$, EMA parameters $\theta_t^*$ can be represented by normally parameter $\theta$ as:

$$\theta_t^* = \prod_{i=1}^{t} \beta_i \cdot \theta_0 + \sum_{i=1}^{t}(1 - \beta_i) \cdot \prod_{j=i+1}^{t} \beta_j \cdot \theta_i. \tag{21}$$

### A.2 PROOF OF RELATIONSHIP BETWEEN $\theta_t$, $\theta_t^*$ AND $\Delta\theta$

From *s.t.* constraint, we have:

$$\Delta\theta = \theta_t^* - \theta_{t-1}^*, \tag{22}$$

$$\theta_{t-1}^* = \theta_t^* - \Delta\theta. \tag{23}$$

Replace $\theta_{t-1}^*$ with $\theta_t^* - \Delta\theta$ in Eq.(1):

$$\theta_t^* = \beta_t(\theta_t^* - \Delta\theta) + (1 - \beta_t)\theta_t. \tag{24}$$

Rearrange the above equation and have:

$$\theta_t^* - \theta_t = \beta_t(\theta_t^* - \theta_t) - \beta_t \Delta\theta, \tag{25}$$

$$(1 - \beta_t)(\theta_t^* - \theta_t) = -\beta_t \Delta\theta. \tag{26}$$

Finally, we can achieve that:

$$\theta_t^* - \theta_t = -\frac{\beta_t}{1 - \beta_t}\Delta\theta = \frac{\beta_t}{\beta_t - 1}\Delta\theta. \tag{27}$$

## A.3  $\beta_t$ Solving Process

With introducing Eq.(23) and Eq.(6), we can represent $\theta_{t-1}^* - \theta_t$ as:

$$\theta_{t-1}^* - \theta_t = \theta_t^* - \Delta\theta - \theta_t = \frac{\beta_t}{\beta_t - 1}\Delta\theta - \Delta\theta = \frac{\Delta\theta}{\beta_t - 1}. \tag{28}$$

Taking the derivative of the Lagrangian to $\Delta\theta$ and setting it to zero as Eq.(11), we have:

$$\frac{\partial F}{\partial \Delta\theta} = \frac{\beta}{(\beta - 1)}\mathcal{L}'(\theta_t) + \frac{\beta^2}{(\beta - 1)^2}\mathcal{L}''(\theta_t)\Delta\theta + 1 + \lambda = 0. \tag{29}$$

Further, we substitute Eq.(28) and Eq.(29) into Eq.(11), and have:

$$0 = -\frac{1}{(\beta_t - 1)^2}\mathcal{L}'(\theta_t)\Delta\theta - \frac{\beta_t}{(\beta_t - 1)^3}\mathcal{L}''(\theta_t)\Delta\theta^2 - [-\frac{\beta_t}{(\beta_t - 1)}\mathcal{L}'(\theta_t) -$$
$$\frac{\beta_t^2}{(\beta_t - 1)^2}\mathcal{L}''(\theta_t)\Delta\theta - 1](\theta_{t-1}^* - \theta_t), \tag{30}$$

$$0 = -\frac{1}{(\beta_t - 1)^2}\mathcal{L}'(\theta_t)\Delta\theta - \frac{\beta_t}{(\beta_t - 1)^3}\mathcal{L}''(\theta_t)\Delta\theta^2 + \frac{\beta_t}{(\beta_t - 1)^2}\mathcal{L}'(\theta_t)\Delta\theta +$$
$$\frac{\beta_t^2}{(\beta_t - 1)^3}\mathcal{L}''(\theta_t)\Delta\theta^2 + \frac{\Delta\theta}{\beta_t - 1}, \tag{31}$$

$$0 = \frac{-1 + \beta_t}{(\beta_t - 1)^2}\mathcal{L}'(\theta_t)\Delta\theta + \frac{-\beta_t + \beta_t^2}{(\beta_t - 1)^3}\mathcal{L}''(\theta_t)\Delta\theta^2 + \frac{\Delta\theta}{\beta_t - 1}, \tag{32}$$

$$0 = \frac{1}{(\beta_t - 1)}\mathcal{L}'(\theta_t)\Delta\theta + \frac{\beta_t}{(\beta_t - 1)^2}\mathcal{L}''(\theta_t)\Delta\theta^2 + \frac{\Delta\theta}{\beta_t - 1}, \tag{33}$$

$$0 = \Delta\theta[\frac{1}{(\beta_t - 1)}\mathcal{L}'(\theta_t) + \frac{\beta_t}{(\beta_t - 1)^2}\mathcal{L}''(\theta_t)\Delta\theta + \frac{1}{\beta_t - 1}]. \tag{34}$$

By observation, we can find one solution that $\Delta\theta = 0$, which means that $\theta_t^* = \theta_{t-1}^*$ and $\beta_t = 1$. Obviously, it is not the global optimal solution due to the lack of updates to EMA parameters.

Then, we can find another solution through the following equation:

$$0 = \frac{1}{(\beta_t - 1)}\mathcal{L}'(\theta_t) + \frac{\beta_t}{(\beta_t - 1)^2}\mathcal{L}''(\theta_t)\Delta\theta + \frac{1}{\beta_t - 1}. \tag{35}$$

Due to the situation that $\beta_t - 1 = 0$ has been discussed, we can remove it unlimited:

$$0 = \mathcal{L}'(\theta_t) + \frac{\beta_t}{(\beta_t - 1)}\mathcal{L}''(\theta_t)\Delta\theta + 1. \tag{36}$$

From Eq.(28), we can get:

$$\Delta\theta = (\theta_{t-1}^* - \theta_t)(\beta_t - 1). \tag{37}$$

Substitute 37 into Eq.(36):

$$0 = \mathcal{L}'(\theta_t) + \beta_t(\theta_{t-1}^* - \theta_t)\mathcal{L}''(\theta_t) + 1. \tag{38}$$

Finally, we obtain another solution for $\beta_t$ that:

$$\beta_t = \frac{\mathcal{L}'(\theta_t) + 1}{(\theta_t - \theta_{t-1}^*)\mathcal{L}''(\theta_t)}. \tag{39}$$

### A.3.1  Discussions of $\beta_t$

**I. Satisfy the *s.t.* equation**

According to the Eq.(28), we have already proved:

$$(\theta_{t-1}^* - \theta_t) = \frac{\Delta\theta}{\beta_t - 1}, \tag{40}$$

$$(\theta_{t-1}^* - \theta_t)(\beta_t - 1) = \Delta\theta. \tag{41}$$

Thus, we can achieve the *i.e.* constraint with the solution as:

$$\Delta\theta + \theta_{t-1}^* - \theta_t^* = (\theta_{t-1}^* - \theta_t)(\beta_t - 1) + \theta_{t-1}^* - \theta_t^* = (\theta_{t-1}^* - \theta_t)\beta_t + \theta_t - \theta_t^*$$
$$= (\theta_{t-1}^* - \theta_t)\beta_t + \theta_t - [\beta_t\theta_{t-1}^* + (1 - \beta_t)\theta_t]$$
$$= \theta_{t-1}^*\beta_t - \theta_t\beta_t + \theta_t - \beta_t\theta_{t-1}^* - \theta_t + \beta_t\theta_t = 0. \tag{42}$$

## A.4 CASES OF MULTIPLE ROUNDS OF DIALOGUE

In this section, we test the zero-shot performance of MLLMs continually fine-tuned with our method on the multiple rounds of dialogue tasks. Images and questions are from (Liu et al., 2024b). To have a comparison, we also test the zero-shot performance of MLLMs continually fine-tuned with the baseline on the multiple rounds of dialogue tasks.

---

**Visual input example, Extreme Ironing:**

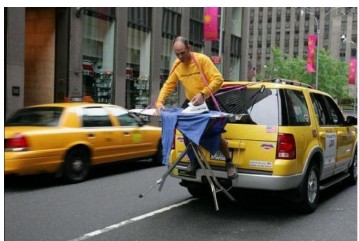

---

**llava-lora (baseline)**

**Human:** Can you describe the image?
**Assistant:** In the image, a man is standing on the back of a yellow truck, **holding a clothesline** with clothes hanging on it. He appears to be **washing clothes** in the middle of a busy street. The truck is driving down the road, and there are other vehicles around, including a taxi and a car.

There are also **several traffic lights visible in the scene**, indicating that the area is likely a busy intersection. The man is focused on his task, despite the bustling environment around him.

**Human:** Do you think there is anything that goes against human intuition?
**Assistant:** Yes, the image of a man **washing clothes** in the middle of a busy street goes against human intuition. Typically, people would expect to see someone **washing clothes** in a more private or secluded area, such as a backyard or a laundromat. However, in this case, the man has chosen to **wash clothes** in the middle of a busy street, which is an unusual and unconventional location for such an activity.

---

**llaca-ours**

**Human:** Can you describe the image?
**Assistant:** In the image, a man is standing on a folding chair in the back of a yellow taxi cab, **ironing clothes**. The taxi is driving down a busy city street, surrounded by other vehicles such as a car and a truck. The man appears to be focused on his task, possibly trying to **dry his clothes quickly** in the moving vehicle.

**Human:** Do you think there is anything that goes against human intuition?
**Assistant:** Yes, the image of a man **ironing clothes** in the back of a taxi cab is unusual and goes against human intuition. It is not a typical scenario one would expect to see, as ironing clothes is usually done in a more comfortable and controlled environment, such as a home or a laundry room. The man's decision to **iron clothes** in the back of a taxi cab suggests that he is **trying to save time or that he might be in a hurry**.

---

Figure 6: Case I of multiple rounds of dialogue, red marks the hallucinations existing in the sentences generated by the baseline, and blue marks that our method can overcome the occurrence of hallucinations in relevant statements and generate correct answers. Besides that, we also discover that our method can provide reasonable explanations for some unusual behaviors.

**Visual input example, Funny Image:**

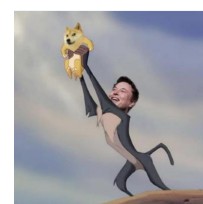

**llava-lora (baseline)**

**Human:** What is the name of the man in the photo?
**Assistant:** the man in the photo is named elon musk
**Human:** Where is the interesting aspect of this picture?
**Assistant:** the interesting aspect of this picture is the man holding a dog
**Human:** What is Elon Musk holding in his hand?
**Assistant:** elon musk is holding a dog
**Human:** What kind of dog is in elon's hand?
**Assistant:** a puppy

**llaca-ours**

**Human:** What is the name of the man in the photo?
**Assistant:** Elon Musk
**Human:** Where is the interesting aspect of this picture?
**Assistant:** Elon Musk's face
**Human:** What is Elon Musk holding in his hand?
**Assistant:** puppy
**Human:** What kind of puppy is in Elon's hand?
**Assistant:** Shiba Inu

Figure 7: Case II of multiple rounds of dialogue, red marks the rough and low-quality answers generated by the baseline, and blue marks that our method can generate fine-grained and high-quality answers. We believe that this skill is learned from the downstream datasets. Besides that, we also discover the generated answers of our method are more concise and to the point, directly addressing the problem.

864
865
866

**Visual input example, Cooking Recipe:**

867
868
869
870
871
872
873

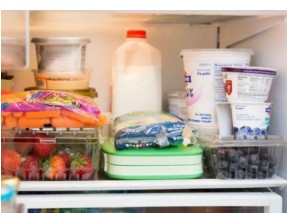

874

**llava-lora (baseline)**

875

**Human:** What are the meals that I can cook with these?
**Assistant:** With the items in the refrigerator, you can prepare a variety of meals. For example, you can make a fruit salad using the strawberries and blueberries. You can also use the carrots to prepare a healthy vegetable stir-fry, a soup, or a salad. The milk can be used to make a smoothie, a sauce, or a creamy soup. The eggs can be used to make a fried egg, an omelette, or a baked dish. Additionally, you can use the cheese to make a pizza, a grilled cheese sandwich, or a cheese-based sauce. The yogurt can be used to make a yogurt parfait, a dip, or a sauce. The butter can be used to cook or bake various dishes, such as bread or pastries. The ice cream can be used to make a dessert or a topping for other dishes.
**Human:** I would like to do the fruit salad. Show me the recipe.
**Assistant:** Sure, here's a simple recipe for a fruit salad:
Ingredients:
* 1 cup strawberries, hulled and sliced
* 1 cup blueberries, rinsed and drained
* 1/2 cup **raspberries**, rinsed and drained
* 1/2 cup **grapes**, seedless
* 1/2 cup **cantaloupe**, diced
* 1/4 cup honey or sugar (optional)
* 1/4 cup lemon juice (optional)
* 1/4 cup chopped fresh mint leaves (optional)
Instructions:
1. In a large bowl, combine the strawberries, blueberries, raspberries, grapes, and cantaloupe.
2. If desired, add honey or sugar to taste.
3. If desired, add lemon juice to the bowl for a tangy flavor.
4. Gently mix the fruit together.
5. Garnish with chopped fresh mint leaves, if desired.
Enjoy your fruit salad!

**llaca-ours**

**Human:** What are the meals that I can cook with these?
**Assistant:** With the variety of fruits, vegetables, and milk available in the refrigerator, you can prepare a range of healthy and delicious meals. Some ideas include:
**1. Fruit salad:** Combine the strawberries, blueberries, and any other fruits available in the refrigerator. You can add a squeeze of lemon juice and a sprinkle of sugar for a simple, refreshing dessert.
**2. Vegetable stir-fry:** Chop up the carrots and any other vegetables available in the refrigerator. Heat some oil in a pan and stir-fry the vegetables with your choice of protein (e.g., tofu, chicken, or beef) and a flavorful sauce.
**3. Smoothie:** Blend the milk with the blueberries, strawberries, and any other fruits available in the refrigerator. You can also add some ice and a scoop of protein powder for a nutritious and delicious smoothie.
**4. Vegetable soup:** Use the carrots and any other vegetables available in the refrigerator to make a vegetable soup. You can add some milk or a vegetable broth for added flavor and nutrition.
**5. Omelette:** Whisk eggs with milk and season with salt and pepper. Cook the omelette in a pan and add the carrots and any other vegetables available in the refrigerator for a nutritious and filling breakfast.
These are just a few examples of the many meals you can prepare using the ingredients available in the refrigerator.
**Human:** I would like to do the fruit salad. Show me the recipe.
**Assistant:** Sure, here's a simple recipe for a fruit salad using the ingredients available in the refrigerator:
Ingredients:
* Strawberries
* Blueberries
* Carrots (optional)
* Milk
* Lemon juice (optional)
* Sugar (optional)
Instructions:
1. Wash the strawberries and blueberries thoroughly.
2. Chop the strawberries into bite-sized pieces.
3. Wash the blueberries and chop them into bite-sized pieces.
4. If you'd like to add carrots, wash and chop them as well.
5. In a large bowl, combine the chopped strawberries, blueberries, and carrots (if using).
6. Pour the milk over the fruit mixture and stir gently.
7. If you'd like to add a little sweetness, squeeze a lemon wedge over the fruit mixture and sprinkle with sugar.
8. Serve the fruit salad immediately or refrigerate for a few hours to allow the flavors to meld together.
Enjoy your delicious and refreshing fruit salad!

Figure 8: Case III of multiple rounds of dialogue, red marks the hallucinations existing in the sentences generated by the baseline, which have not existed in the image. Blue marks that our method can generate answers that are clearer and more aligned with human reading habits.

**Visual input example, Webpage Making:**

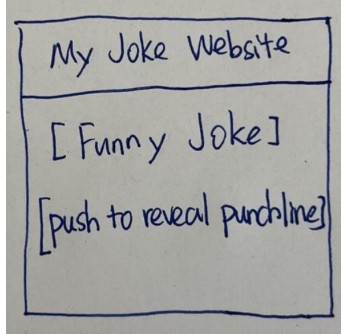

**llava-lora (baseline)**

**Human:** Please turn this mock-up into an interactive website using html/js, and replace the joke with a real joke.
**Assistant:** I'm sorry, but I am not able to create an interactive website using html/js. However, I can provide you with a real joke:

Why was the math book sad?

Because it had too many problems.

**llaca-ours**

**Human:** Please turn this mock-up into an interactive website using html/js, and replace the joke with a real joke.
**Assistant:** Sure, I can help you with that! Here's a simple HTML/JS mock-up of the joke website:
```html
<!DOCTYPE html>
<html>
 <head>
   <title>My Joke Website</title>
 </head>
 <body>
   <h1>My Joke Website</h1>
   <h2>Funny Joke</h2>
   <p>Push to reveal punchline</p>
   <button onclick="revealPunchline()">Reveal Punchline</button>
   <p id="punchline"></p>
   <script>
    function revealPunchline() {
      document.getElementById("punchline").innerHTML = "Why did the tomato turn red?";
    }
   </script>
 </body>
</html>
```
This is a simple mock-up of the joke website, which includes a button to reveal the punchline. The punchline is stored in the `punchline` element, which is hidden by default. When the button is clicked, the `revealPunchline()` function is called, which sets the text of the `punchline` element to "Why did the tomato turn red?".

Figure 9: Case IV of multiple rounds of dialogue, after continually fine-tuning, the baseline has forgotten the pre-trained knowledge and lost the ability to write HTML/JS code. While our method can protect the previous knowledge and still own the ability of webpage code writing after continually fine-tuning.

## A.5 DETAILED EXPERIMENTAL RESULTS WITH DIFFERENT TRAINING STRATEGIES

Table 6: Continual tuning performance of LLaCA after each dataset. Results are obtained with training strategy type 1.

| ScienceQA | TextVQA | GQA | VizWiz | VQAv2 | OCRVQA |
|---|---|---|---|---|---|
| 80.19 | | | | | |
| 78.24 | 59.69 | | | | |
| 77.74 | 57.96 | 61.58 | | | |
| 78.00 | 58.34 | 60.52 | 52.00 | | |
| 76.66 | 58.10 | 60.00 | 36.47 | 66.78 | |
| 77.15 | 56.54 | 60.18 | 47.16 | 65.83 | 64.45 |

Table 7: Continual tuning performance of LLaCA after each dataset. Results are obtained with training strategy type 2.

| ScienceQA | TextVQA | GQA | VizWiz | VQAv2 | OCRVQA |
|---|---|---|---|---|---|
| 80.92 | | | | | |
| 79.18 | 59.37 | | | | |
| 78.47 | 57.88 | 61.85 | | | |
| 78.09 | 58.51 | 60.46 | 50.29 | | |
| 78.07 | 58.56 | 59.83 | 36.68 | 66.57 | |
| 77.91 | 57.15 | 60.38 | 40.19 | 66.48 | 65.27 |

Table 8: Continual tuning performance of LLaCA after each dataset. Results are obtained with training strategy type 3.

| OCRVQA | VQAv2 | VizWiz | GQA | TextVQA | ScienceQA |
|---|---|---|---|---|---|
| 64.84 | | | | | |
| 64.79 | 67.24 | | | | |
| 63.42 | 66.93 | 48.88 | | | |
| 62.79 | 65.16 | 49.78 | 62.18 | | |
| 63.32 | 65.94 | 42.60 | 60.91 | 60.10 | |
| 62.48 | 65.77 | 47.33 | 60.98 | 59.18 | 74.94 |

Table 9: Continual tuning performance of LLaCA after each dataset. Results are obtained with training strategy type 4.

| GQA | OCRVQA | ScienceQA | VQAv2 | TextVQA | VizWiz |
|---|---|---|---|---|---|
| 60.46 | | | | | |
| 60.21 | 64.15 | | | | |
| 59.68 | 61.73 | 73.71 | | | |
| 59.91 | 57.00 | 72.58 | 66.37 | | |
| 60.75 | 59.86 | 72.37 | 66.57 | 59.80 | |
| 60.20 | 59.87 | 72.15 | 66.59 | 58.10 | 49.39 |

## A.6 COMPARED METHODS

**LoRA (Hu et al., 2022)** prepends LoRA parameter efficient tuning paradigm into LLM. In the training stage, it only trains the linear projector and LoRA parameters, with frozen vision encoder and LLM; **MoELoRA (Chen et al., 2024a)** is based on the LoRA, and the number of experts for each MoE layer is set to 2; **LWF (Li & Hoiem, 2017)** calculates the results of the new dataset samples on both the old and new models. After that, it calculates the distillation loss and adds it to the loss function as a regularization penalty term. **EWC (Kirkpatrick et al., 2017)** considers the change of the training parameters and proposes the specific parameters changing loss as a regularization penalty. **PGP (Qiao et al., 2024a)** introduces a gradient projection method for efficient parameters, and changes the gradient direction orthogonal to the previous feature subspace.

## A.7 DETAILED IMPLEMENTATION

Based on Eq.(14), we continue to further simplify it as:

$$
\begin{aligned}
\beta_t &\approx \| \frac{[\mathcal{L}'(\hat{\theta}_t) + 1](\hat{\theta}_t - \hat{\theta}_{t-1})}{(\hat{\theta}_t - \hat{\theta}_{t-1}^*)[\mathcal{L}'(\hat{\theta}_t) - \mathcal{L}'(\hat{\theta}_{t-1})]} \| \\
&= \| \frac{[\mathcal{L}'(\hat{\theta}_t) + 1](\hat{\theta}_t - \hat{\theta}_{t-1}^* + \hat{\theta}_{t-1}^* - \hat{\theta}_{t-1})}{(\hat{\theta}_t - \hat{\theta}_{t-1}^*)[\mathcal{L}'(\hat{\theta}_t) - \mathcal{L}'(\hat{\theta}_{t-1})]} \| \\
&= \| \frac{\mathcal{L}'(\hat{\theta}_t) + 1}{\mathcal{L}'(\hat{\theta}_t) - \mathcal{L}'(\hat{\theta}_{t-1})} - \frac{[\hat{\theta}_{t-1} - \hat{\theta}_{t-1}^*][\mathcal{L}'(\hat{\theta}_t) + 1]}{(\hat{\theta}_t - \hat{\theta}_{t-1}^*)[\mathcal{L}'(\hat{\theta}_t) - \mathcal{L}'(\hat{\theta}_{t-1})]} \| \\
&= \| \frac{\mathcal{L}'(\hat{\theta}_t) + 1 - \mathcal{L}'(\hat{\theta}_{t-1}) + \mathcal{L}'(\hat{\theta}_{t-1})}{\mathcal{L}'(\hat{\theta}_t) - \mathcal{L}'(\hat{\theta}_{t-1})} - \frac{[\hat{\theta}_{t-1} - \hat{\theta}_{t-1}^*][\mathcal{L}'(\hat{\theta}_t) + 1]}{(\hat{\theta}_t - \hat{\theta}_{t-1}^*)[\mathcal{L}'(\hat{\theta}_t) - \mathcal{L}'(\hat{\theta}_{t-1})]} \| \\
&= \| 1 + \frac{1 + \mathcal{L}'(\hat{\theta}_{t-1})}{\mathcal{L}'(\hat{\theta}_t) - \mathcal{L}'(\hat{\theta}_{t-1})} - \frac{[\hat{\theta}_{t-1} - \hat{\theta}_{t-1}^*][\mathcal{L}'(\hat{\theta}_t) + 1]}{(\hat{\theta}_t - \hat{\theta}_{t-1}^*)[\mathcal{L}'(\hat{\theta}_t) - \mathcal{L}'(\hat{\theta}_{t-1})]} \|.
\end{aligned}
\tag{43}
$$

Additionally, by observation in experiments, we find that $\|\mathcal{L}'(\hat{\theta}_{t-1}) + 1\| \ll \|\mathcal{L}'(\hat{\theta}_t) - \mathcal{L}'(\hat{\theta}_{t-1})\|$, leading to:

$$
\| \frac{1 + \mathcal{L}'(\hat{\theta}_{t-1})}{\mathcal{L}'(\hat{\theta}_t) - \mathcal{L}'(\hat{\theta}_{t-1})} \| \approx 0.
\tag{44}
$$

Therefore, Eq.(43) could be transfered as:

$$
\beta_t \approx \| 1 - \frac{[\hat{\theta}_{t-1} - \hat{\theta}_{t-1}^*][\mathcal{L}'(\hat{\theta}_t) + 1]}{(\hat{\theta}_t - \hat{\theta}_{t-1}^*)[\mathcal{L}'(\hat{\theta}_t) - \mathcal{L}'(\hat{\theta}_{t-1})]} \|.
\tag{45}
$$

The above is our final result, and we approximate $\beta_t$ using the Eq.(45) in implementation.

## A.8 TRAINING DETAILS

In the implementation of our method, the codebase is based on CoIN (Chen et al., 2024a) and LLaVA (Liu et al., 2024a). The vision tower is clip-vit-large-patch14-336 pre-trained by OpenAI and the LLM is Vicuna-7B. The inserted LoRA in each module layer of LLM has a rank of 128. For each fine-tuning dataset, the training epoch is set to 1, and the initial learning rate and weight decay are configured at 2e-4 and 0. The max length of input text is fitted as 2048. Additionally, we adopt gradient checkpoint strategy and mixed precision mode of TF32 and BF16. Furthermore, we also utilize the ZeRO stage: 0 mode of DeepSpeed for training. All experiments are conducted on 8 NVIDIA A100 GPUs with 80GB of memory, excluding Table 3.

## A.9 EVALUATION METRICS

It is worth noting that our judgment of whether the prediction results are correct or not is strictly based on the direct comparison between outputs of MLLMs and ground truths, which is defined as **Instruction Following Ability** in Chen et al. (2024a). Therefore, our judgment criteria would be more stringent.

**Average Accuracy** (Avg.ACC) is used for averaging the test accuracy of all datasets, which represents the comprehensive performance of continual tuning.

**Forgetting** (FOR) is utilized to indicate the test accuracy reduction of past datasets after learning the new dataset, which denotes the stability performance.

**New Accuracy** (New.ACC) is employed to average the test accuracy of new datasets, which refers to the plasticity performance.

**Average Dataset Accuracy** (ADA) refers to the average accuracy of a specific dataset in its current training dataset and the following datasets.

**Average Dataset Forgetting** (ADF) refers to the average forgetting of a specific dataset in its current training dataset and the following datasets.

(1). Average Accuracy, Forgetting, and New Accuracy are generally defined as:

$$\text{Average Accuracy} = \frac{1}{T}\sum_{i=1}^{T} A_{T,i}, \tag{46}$$

$$\text{Forgetting} = \frac{1}{T-1}\sum_{i=1}^{T-1} A_{T,i} - \max(A_{j,i})_{j\in[i,T-1]}, \tag{47}$$

$$\text{New Accuracy} = \frac{1}{T}\sum_{i=1}^{T} A_{i,i}, \tag{48}$$

where $T$ is the number of datasets, $A_{T,i}$ is the accuracy of $i$-th dataset on the model trained after $T$-th dataset, $A_{j,i}$ is the accuracy of $i$-th dataset on the model trained after $j$-th dataset, and $A_{i,i}$ is the accuracy of $i$-th dataset on the model trained after $i$-th dataset.

(2). Average Dataset Accuracy and Average Dataset Forgetting are generally defined as:

$$\text{Average Dataset Accuracy} = \frac{1}{T-t+1}\sum_{i=t}^{T} A_{i,t}, \tag{49}$$

$$\text{Average Dataset Forgetting} = \frac{1}{T-t}\sum_{i=t+1}^{T} F_{i,t}, \tag{50}$$

where $T$ is the number of datasets, $t$ is the specific dataset, $A_{i,t}$ is the accuracy of $t$-th dataset on the model trained after $i$-th dataset, $F_{t,i}$ is the forgetting of $t$-th dataset on the model trained after $i$-th dataset.

### A.10 TYPES OF ROBUST TRAINING STRATEGY

In order to validate the robustness of our method, we design the following four types of training strategy, mixed with distinct instruction types and various training orders.

1). Instruction type 1 and training order: ScienceQA, TextVQA, GQA, VizWiz, VQAv2, OCRVQA.

2). Instruction type 2 and training order: ScienceQA, TextVQA, GQA, VizWiz, VQAv2, OCRVQA.

3). Instruction type 1 and training order: OCRVQA, VQAv2, VizWiz, GQA, TextVQA, ScienceQA.

4). Instruction type 2 and training order: GQA, OCRVQA, ScienceQA, VQAv2, TextVQA, VizWiz.

### A.11 DISTRIBUTION OF $\beta_t$ IN DISTINCT DATASETS

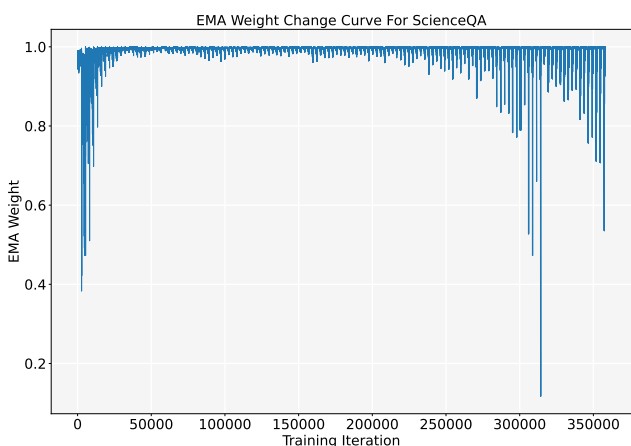

Figure 10: Distribution of $\beta_t$ in training ScienceQA dataset with Instruction type1. The horizontal axis represents iteration, and the vertical axis represents value of $\beta_t$.

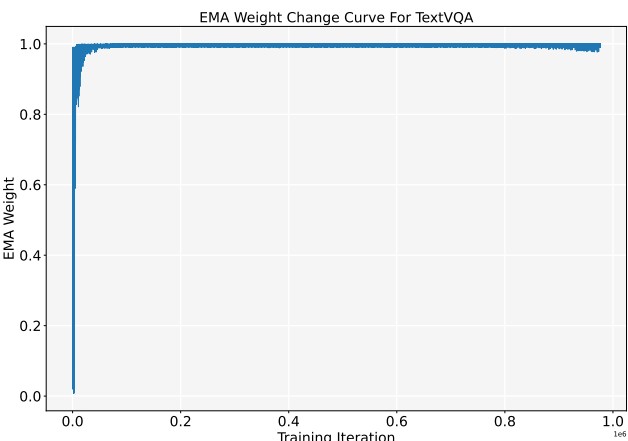

Figure 11: Distribution of $\beta_t$ in training TextVQA dataset with Instruction type1. The horizontal axis represents iteration, and the vertical axis represents value of $\beta_t$.

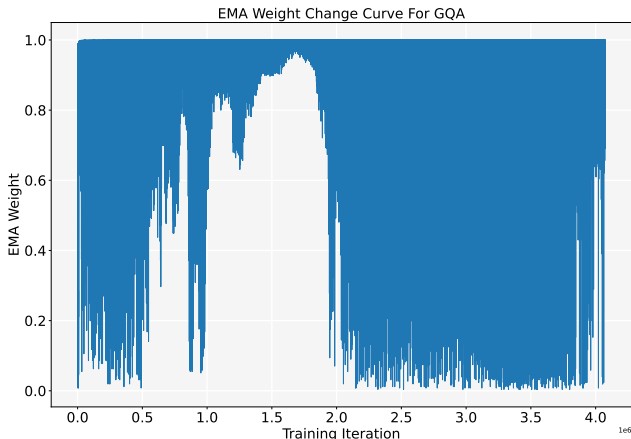

Figure 12: Distribution of $\beta_t$ in training GQA dataset with Instruction type1. The horizontal axis represents iteration, and the vertical axis represents value of $\beta_t$.

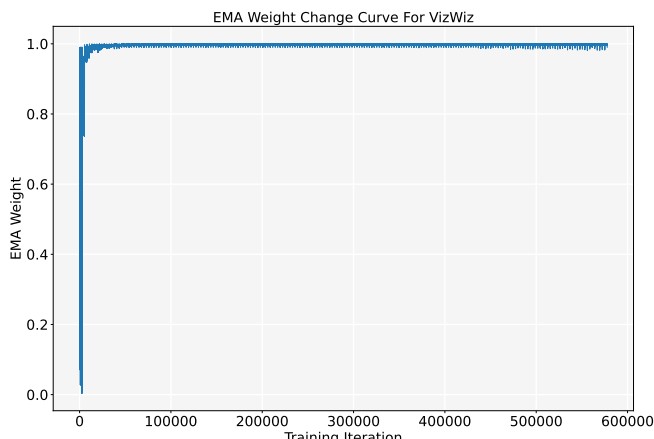

Figure 13: Distribution of $\beta_t$ in training VizWiz dataset with Instruction type1. The horizontal axis represents iteration, and the vertical axis represents value of $\beta_t$.

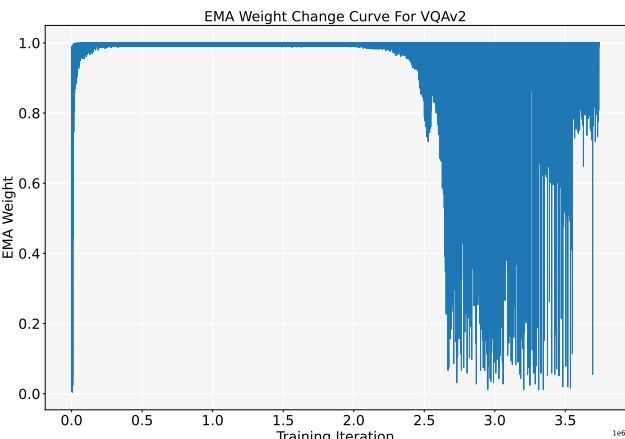

Figure 14: Distribution of $\beta_t$ in training VQAv2 dataset with Instruction type1. The horizontal axis represents iteration, and the vertical axis represents value of $\beta_t$.

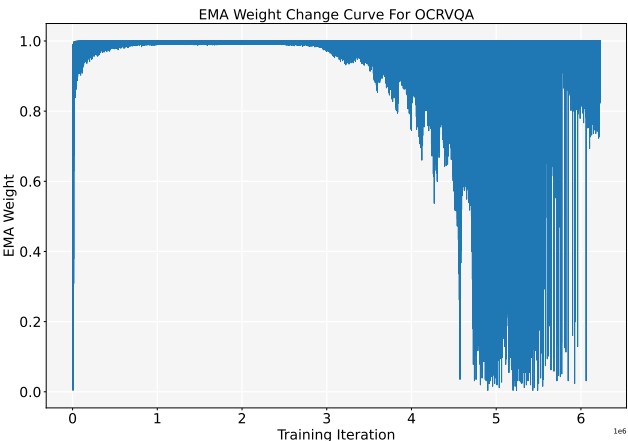

Figure 15: Distribution of $\beta_t$ in training OCRVQA dataset with Instruction type1. The horizontal axis represents iteration, and the vertical axis represents value of $\beta_t$.

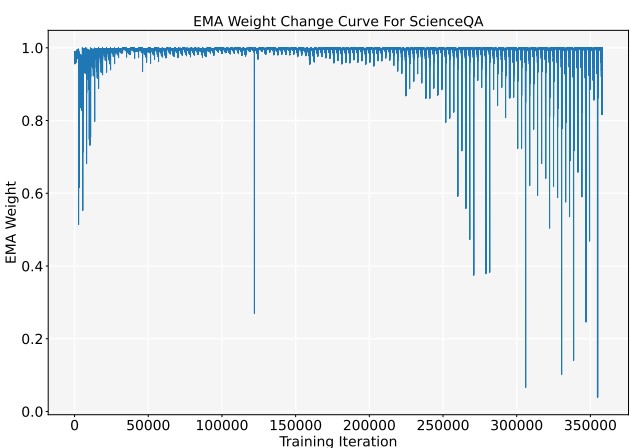

Figure 16: Distribution of $\beta_t$ in training ScienceQA dataset with Instruction type2. The horizontal axis represents iteration, and the vertical axis represents value of $\beta_t$.

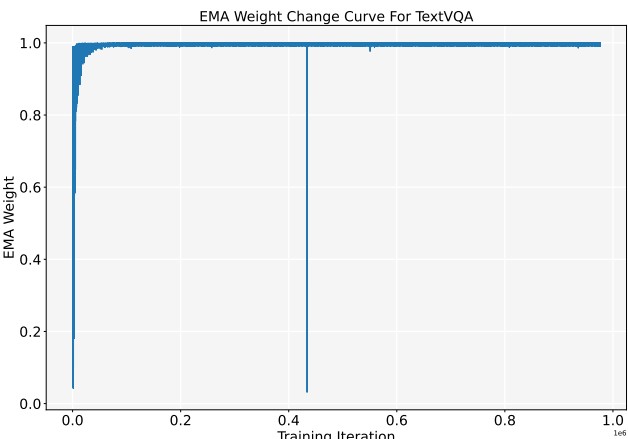

Figure 17: Distribution of $\beta_t$ in training TextVQA dataset with Instruction type2. The horizontal axis represents iteration, and the vertical axis represents value of $\beta_t$.

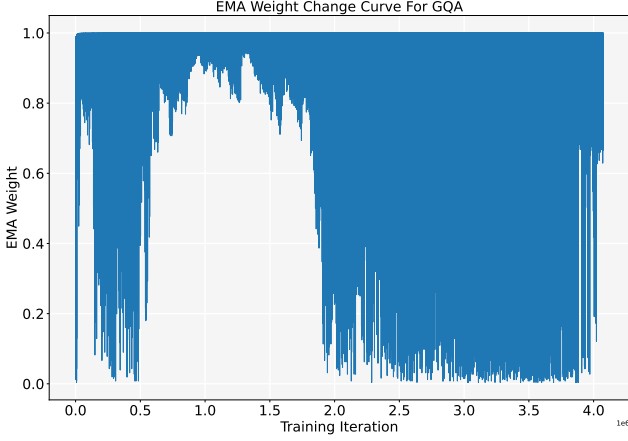

Figure 18: Distribution of $\beta_t$ in training GQA dataset with Instruction type2. The horizontal axis represents iteration, and the vertical axis represents value of $\beta_t$.

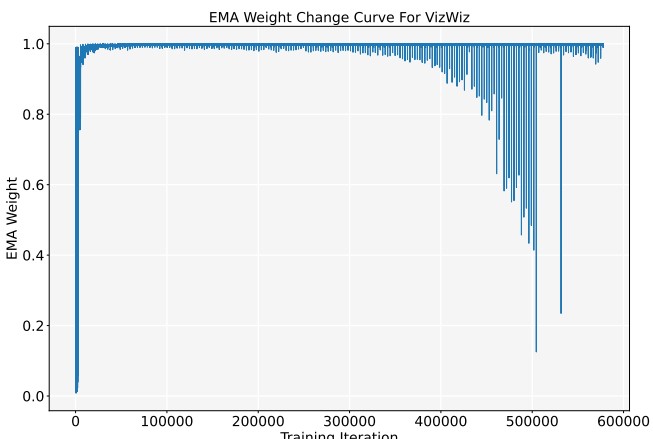

Figure 19: Distribution of $\beta_t$ in training VizWiz dataset with Instruction type2. The horizontal axis represents iteration, and the vertical axis represents value of $\beta_t$.

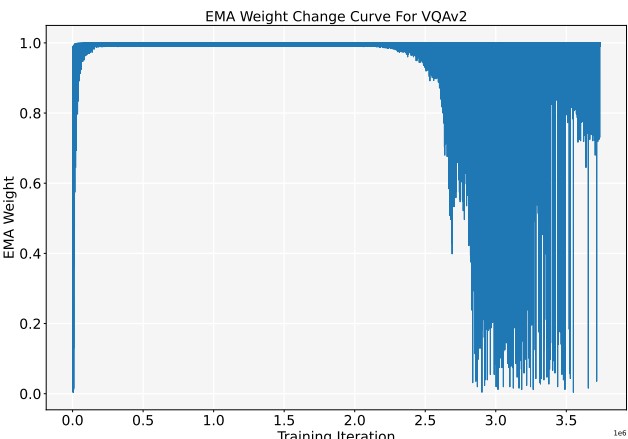

Figure 20: Distribution of $\beta_t$ in training VQAv2 dataset with Instruction type2. The horizontal axis represents iteration, and the vertical axis represents value of $\beta_t$.

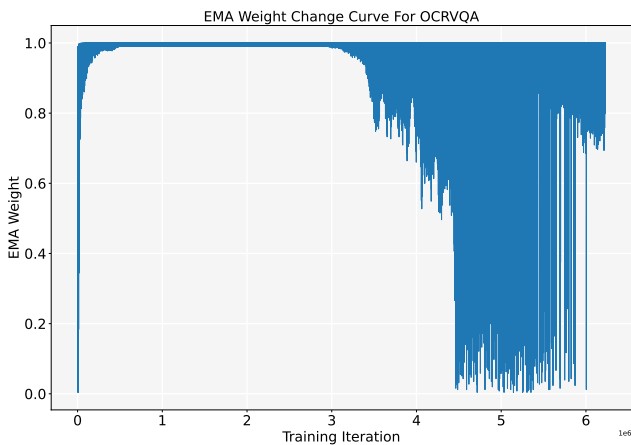

Figure 21: Distribution of $\beta_t$ in training OCRVQA dataset with Instruction type2. The horizontal axis represents iteration, and the vertical axis represents value of $\beta_t$.

## A.12 ALGORITHM OF LLaCA

---
**Algorithm 1:** Multimodal Large Language Continual Assistant (Training phase)

---
**Input:** Pre-trained ViT model $f_{vis}$, Pre-trained Vicuna-7B model $f_{lan}$ with inserted LoRA
$f_{low}$, projection layer $f_{prj}$, embedding layer $\phi$, number of datasets $D$, number of
iterations $T$, training set $\{\{G_i^t, T_i^t, I_i^t, y_i^t\}_{i=1}^{n_t}\}_{t=1}^T$, learning rate $\eta$, loss function $\mathcal{L}_x$.
**Output:** inserted LoRA $f_{low}^*$, projection layer $f_{prj}^*$.
**initialize:** $f_{low}, f_{low}^*$ $f_{prj}, f_{prj}^*$.
**for** $d = 1, ..., D$ **do**
    **for** $epoch = 1$ **do**
        **for** $t = 1, ..., T$ **do**
            1.Draw a mini-batch $B = \{(G_i^t, T_i^t, I_i^t, y_i^t\}_{i=1}^{n_t})\}_{i=1}^{n_t}$.
            **for** $(G, T, I, y)$ *in* $B$ **do**
                2.Encode $G$ into image feature $g_t$ by $g_t = f_{vis}(G)$.
                3.Project $g_t$ from image feature space to text feature space as $p_t = f_{prj}(g_t)$.
                4.Embed $T$ and $I$ into text feature $e_t$ by $e_t = \phi([T, I])$.
                5.Prepend $p_t$ with $e_t$ by $[p_t; e_t]$.
                6.Obtain prediction by $\hat{y} = f_{lan}([p_t; e_t])$.
                7.Calculate per batch loss $\mathcal{L}_B$ by accumulating $\mathcal{L}_x(y, \hat{y})$.
                8.Backward propagation and Update $f_{low}$ and $f_{prj}$ with optimizer.
                9.Record $f_{low}, f_{prj}$ and the corresponding gradients $\mathcal{L}'$ at iteration $t$.
                # EMA weight calculate.
                10.Calculate EMA weight $\beta_t$ according to Eq.(45).
                # EMA parameter update.
                11.Update $f_{low}^*$ and $f_{prj}^*$ by Eq.(1).
                12.Remove and clear $f_{low}, f_{prj}$ and $\mathcal{L}'$ at iteration $t - 1$.
            **end**
        **end**
    **end**
    13. Save checkpoints of $f_{low}^*$ and $f_{prj}^*$.
**end**

---

---
**Algorithm 2:** Multimodal Large Language Continual Assistant (Testing phase)

---
**Input:** Pre-trained ViT model $f_{vis}$, Pre-trained Vicuna-7B model $f_{lan}$ with inserted LoRA
$f_{low}$, projection layer $f_{prj}$, embedding layer $\phi$, number of datasets $D$, number of
iterations $T$, test set $\{\{G_i^t, T_i^t, I_i^t, y_i^t\}_{i=1}^{n_t}\}_{t=1}^T$.
**Output:** prediction $\hat{y}$.
**for** $d = 1, ..., D$ **do**
    **for** $t = 1, ..., T$ **do**
        1.Draw a mini-batch $B = \{(G_i^t, T_i^t, I_i^t\}_{i=1}^{n_t})\}_{i=1}^{n_t}$.
        **for** $(G, T, I)$ *in* $B$ **do**
            2.Encode $G$ into image feature $g_t$ by $g_t = f_{vis}(G)$.
            3.Project $g_t$ from image feature space to text feature space as $p_t = f_{prj}(g_t)$.
            4.Embed $T$ and $I$ into text feature $e_t$ by $e_t = \phi([T, I])$.
            5.Prepend $p_t$ with $e_t$ by $[p_t; e_t]$.
            6.Obtain prediction by $\hat{y} = f_{lan}([p_t; e_t])$.
        **end**
    **end**
**end**

---

