# OpenReview forum: "LLaCA: Multimodal Large Language Continual Assistant"
_ICLR.cc/2025/Conference — Submitted to ICLR 2025_

### Official Review · Reviewer_RGJd · 2024-10-25

**Soundness:** 3
**Presentation:** 3
**Contribution:** 3
**Rating:** 5
**Confidence:** 5

**Summary:**

The paper proposes the Multimodal Large Language Continual Assistant (LLaCA), a method designed to improve the continual instruction tuning of MLLMs. LLaCA focuses on mitigating catastrophic forgetting when fine-tuning MLLMs with new instructions. The method employs a dynamic EMA strategy to balance retaining old knowledge while learning new information. Experimental results show that the approach improves both anti-forgetting and continual tuning performance compared to baseline methods.

**Strengths:**

1. The paper provides detailed mathematical formulations and theoretical derivations.
2. The method demonstrates strong performance compared to the baselines, showing significant improvements in reducing catastrophic forgetting and enhancing continual learning.
3. Compared to basic LoRA fine-tuning, LLaCA adds slight extra computational cost, highlighting its efficiency.

**Weaknesses:**

1. LLaCA appears to be more aligned with general Continual Learning (CL) methods and lacks specific adaptations for Multimodal Continual Instruction Tuning (MCIT), such as addressing the interaction between multiple modalities and managing diverse instruction types across sequential tasks. The paper primarily focuses on the EMA update strategy and dynamically computed $\beta$, but it does not sufficiently address the unique challenges of MCIT. Furthermore, the specific advantages of applying LLaCA in the MCIT context, such as how it enhances multimodal learning, need to be clarified.

2. The experimental setup generally follows the CoIN [1] benchmark but omits the Referring Expression Comprehension (RefCoCo) and Image Classification (ImageNet) datasets without explanation, focusing solely on the Image Question Answering task. For multimodal large language models (MLLMs), these tasks do not differ fundamentally in format from Image Question Answering. Their exclusion raises concerns about the model's performance on these tasks. Including them would provide a more complete evaluation.

**Questions:**

1. As mentioned in Weakness 2, this paper omits 2 tasks from CoIN, so directly copying the full fine-tuning results from CoIN is incorrect. The authors should reproduce the experiment themselves.
2. In Section 5.1, the authors mention that having "ten kinds of instruction templates" (Instruction Type 2) is more challenging. However, in continual learning, different instructions can serve as implicit task identifiers, helping MLLMs find relevant knowledge for specific tasks.
3. The subsection "Multimodal Continual Instruction Tuning Definition" in Section 2 appears to focus more on problem formulation than related work. Moving it to a more suitable section could improve the paper's overall clarity.
4. The "checkpoints" mentioned in Figure 3 and Algorithm 1 are unclear. If checkpoints are being saved, there should be corresponding loading operations. Could the authors clarify this process?
5. The framework caption in Figure 3 could benefit from more detailed explanations of each module and process to improve clarity and help readers better understand the methodology.
6. There are inconsistencies in the implementation details: Appendix A.8 mentions experiments on 8 NVIDIA A100 GPUs, while Table 3 reports training on 8 A6000 GPUs. Could the authors clarify this discrepancy?
7. The term "baseline" is used throughout the paper, but it is unclear whether it consistently refers to LoRA fine-tuning or other methods. Could the authors clarify what "baseline" refers to, particularly in the tables? It would also be helpful to compare the method with stronger baselines like PGP instead of just LoRA (e.g., Table 4).
8. The paper provides detailed results for seen tasks (Appendix A.5), but results for unseen tasks are missing, despite the discussion of zero-shot performance in Section 5.3. Including these results would clarify how training on certain tasks affects zero-shot performance and explain the improvements over the baseline and original LLaVA-v1.5.
9. The paper cites many arXiv preprints, some of which have formal publications. It would be better to reference the peer-reviewed versions where available.
10. The numbering of multi-line equations representing the same derivation in the appendix is inconsistent (e.g., Eqs. (20)-(21) and (23)-(25)). Using a single equation number for each derivation would be clearer.

    **[Reference]**

    [1] Chen C, Zhu J, Luo X, et al. CoIN: A Benchmark of Continual Instruction Tuning for Multimodal Large Language Model. arXiv preprint arXiv:2403.08350, 2024.

---

> ### Author Response · Authors · 2024-11-22
>
> Dear reviewer RGJd, thanks for your valuable suggestions. Here are our responses:
>
> For **Weakness 1** about ''how it enhances multimodal learning'':
>
> Thanks for your valuable suggestion. To show how it addresses the interaction between multiple modalities and further enhances multimodal learning. We conduct the following ablation experiments (conducted under Instruction Type 1): **Normal**. Use the normal gradient update method to update the projection layer from the visual model to the language model and LoRA in LLM. 2): **Dy(Proj)**. Use dynamic EMA update (our method) to update the projection layer from the visual model to the language model, while updating the LoRA in LLM with the normal gradient update method. 3): **Dy(LLM)**. Use dynamic EMA update (our method) to update the LoRA in LLM, while updating the projection layer from the visual model to the language model with the normal gradient update method. 4): **Dy(Proj+LLM)**. Use dynamic EMA update (our method) to update the LoRA in LLM and the projection layer from the visual model to the language model. 5): **Dy(Vis)**. Extra insert LoRA into the vision encoder and use dynamic EMA update (our method) to update the LoRA in the vision encoder, while updating the projection layer from the visual model to the language model, and LoRA in LLM with the normal gradient update method. 6): **Dy(Proj+LLM+Vis)**. Extra insert LoRA into the vision encoder and use dynamic EMA update (our method) to update the LoRA in the vision encoder, LoRA in LLM, and the projection layer from the visual model to the language model. The results are shown as:
>
> |   Metric   | Normal | Dy(Proj) | Dy(LLM) | Dy(Proj+LLM) | Dy(Vis) | Dy(Proj+LLM+Vis) |
> | :--------: | :----: | :------: | :-----: | :----------: | :-----: | :-----------------: |
> |  Avg. ACC  | 41.31  |  55.87   |  61.83  |    61.89     |  50.72  |        62.38        |
> | Forgetting | 22.67  |  11.12   |  2.96   |     2.68     |  13.24  |        2.31         |
> |  New ACC   | 60.20  |  65.13   |  64.30  |    64.12     |  63.85  |        64.31        |
>
> From the above experimental results, we can explore how our method enhances multimodal learning. The analysis is as follows:
>
> (1). When training the projection layer, it is equivalent to **mapping the encoded visual features to the language-text space**. Both the first-order and second-order gradients of the projection layer collect the information from two modalities. Therefore, when calculating the EMA update weight of the projection layer, it can be seen as a result obtained on the basis of **addressing the interaction between multiple modalities**.
>
> (2). Similarly, after concatenating the encoded visual features and the textual features into union features, we input them into LLM. Therefore, **the features processed by LoRA are a mixture of visual and textual features**. Naturally, both the first-order and second-order gradients of LoRA contain information from two modalities. Therefore, when calculating the EMA update weights of the LoRA layer by utilizing the first-order and second-order gradients, it can also be seen as a result obtained on the basis of **addressing the interaction between multiple modalities**.
>
> (3). It is worth mentioning that although LoRA is also inserted into the vision encoder in **Dy(Proj+LLM+Vis)** and dynamically updated according to the proposed method, its performance is slightly better than that in **Dy(Proj+LLM)**. Similarly, in **Dy(Vis)**, we also insert additional LoRA into the Vision Encoder and update it according to the proposed method, while keeping other trainable parameters updated with the normal gradient update method. It is still inferior to the results in **Dy(Proj)** and **Dy(LLM)**. The above experiments all explain that in the case of a single modality (image), our method has little effect and can only achieve maximum benefits in multimodal scenarios. This also reflects that **our method enhances multimodal learning**.

---

> > ### Comment · Reviewer_RGJd · 2024-11-27
> > **Thank the authors for the rebuttal**
> >
> > The authors provide detailed responses to my questions. I appreciate the efforts on the experiments to validate its effectiveness on the LMM. However, it is more like a general method but does not specifically consider multimodal interaction, fusion, or balancing problems. I will maintain my score.

---

> > > ### Author Response · Authors · 2024-11-28
> > > **Thanks for Reviewer RGJd**
> > >
> > > Thank you for your thoughtful review comments. With your valuable comments and suggestions, we have significantly improved the clarity, fluency, and details of our paper. We sincerely appreciate you and your efforts in reviewing our paper.

---

> ### Author Response · Authors · 2024-11-22
>
> For **Weakness 2** about ''Including them would provide a more complete evaluation'':
>
> Thanks for your valuable suggestion, we add the Image Classification Task (ImageNet-1K) and Referring Expression Comprehension (RefCoCo) in the MCIT task, consistently with CoIN [1]. The new tuning sequence order is as follows: ScienceQA->TextVQA->ImageNet->GQA->VizWiz->Grounding ->VQAv2->OCRVQA, with a total of 8 tasks. The following are the experimental results:
>
> Instruction Type 1
>
> | Method  | ScienceQA | TextVQA | ImageNet |  GQA  | VizWiz | Grounding | VQAv2 | OCRVQA | Avg. ACC  | Forgetting |
> | :-----: | :-------: | :-----: | :------: | :---: | :----: | :-------: | :---: | :----: | :-------: | :--------: |
> |  LoRA   |   49.28   |  31.19  |  10.36   | 36.54 | 28.03  |   1.42    | 46.35 | 52.87  |   32.01   |   27.38    |
> | MoELoRA |   63.09   |  38.63  |  10.50   | 37.38 | 43.62  |   0.59    | 43.15 | 60.08  |   37.13   |   25.91    |
> |  Ours   |   75.63   |  54.47  |  43.64   | 60.70 | 43.37  |   36.00   | 65.21 | 63.59  | **55.33** |  **7.04**  |
>
> Instruction Type 2
>
> | Method  | ScienceQA | TextVQA | ImageNet |  GQA  | VizWiz | Grounding | VQAv2 | OCRVQA | Avg. ACC  | Forgetting |
> | :-----: | :-------: | :-----: | :------: | :---: | :----: | :-------: | :---: | :----: | :-------: | :--------: |
> |  LoRA   |   42.17   |  30.25  |   9.68   | 36.41 | 26.22  |   0.56    | 42.93 | 58.19  |   30.80   |   32.76    |
> | MoELoRA |   47.34   |  32.91  |  10.12   | 37.15 | 42.48  |   0.97    | 42.77 | 57.50  |   33.91   |   28.55    |
> |  Ours   |   74.23   |  56.90  |  37.35   | 60.03 | 48.65  |   22.27   | 66.02 | 63.80  | **53.66** |  **9.62**  |
>
> It can be seen that under 8 instruction tuning tasks, our method still has better performances compared to Baseline (Inserted with LoRA) and MOELoRA [1]. It is worth noting that **our judgment of whether the prediction results are correct or not is strictly based on the direct comparison between outputs of MLLMs and ground truths**, which is defined as **Instruction Following Ability** in [1]. Therefore, our judgment criteria would be more stringent. It can be seen that under such strict standards, the baseline model (LLaVA+LoRA) experiences extremely severe catastrophic forgetting, but **our model still has excellent performance**.
>
> [1] COIN: A Benchmark of Continual Instruction Tuning for Multimodel Large Language Model

---

> ### Author Response · Authors · 2024-11-22
>
> For **Question 1** about ''The authors should reproduce the experiment themselves'':
>
> Thanks for your valuable suggestion, we reproduce the experiment according to our training datasets and orders (ScienceQA->TextQA->GQA->VizWiz->VQAv2->OCRVQA). The following are experimental results.
>
> Instruction Type 1
>
> | ScienceQA | TextVQA |  GQA  | VizWiz | VQAv2 | OCRVQA | Avg.ACC | Forgetting | New ACC |
> | :-------: | :-----: | :---: | :----: | :---: | :----: | :-----: | :--------: | :-----: |
> |   25.72   |  30.56  | 38.49 | 34.42  | 44.75 | 58.84  |  38.80  |   26.87    |  61.62  |
>
> Instruction Type 2
>
> | ScienceQA | TextVQA |  GQA  | VizWiz | VQAv2 | OCRVQA | Avg.ACC | Forgetting | New ACC |
> | :-------: | :-----: | :---: | :----: | :---: | :----: | :-----: | :--------: | :-----: |
> |   28.36   |  27.43  | 34.18 | 27.66  | 41.03 | 54.38  |  35.51  |   26.92    |  60.06  |

---

> ### Author Response · Authors · 2024-11-22
>
> For **Question 2** about ''ten kinds of instruction'':
>
> (1). In our setting, **each tuning task includes all ten kinds of instruction templates**. In other words, they are **task-shared**, not task-specific. Thus, **they cannot serve as implicit task identifiers**.
>
> (2). We obtain the conclusion from **an experimental perspective that all performances of almost all methods fell short on Instruction Type 2 compared with results on Instruction Type 1**.

---

> ### Author Response · Authors · 2024-11-22
>
> For **Question 3** about ''Moving it to a more suitable section'':
>
> Thanks for your valuable suggestion, we have moved the subsection "Multimodal Continual Instruction Tuning Definition" in Section 2 to a more suitable location and enhanced the clarity of the paper.

---

> ### Author Response · Authors · 2024-11-22
>
> For **Question 4** about ''The "checkpoints" mentioned in Figure 3 and Algorithm 1 are unclear'':
>
> In fact, we do not load the checkpoints during the training process. Here we just temporarily store the parameter checkpoints in the form of a Parameter Dictionary in Pytorch for calculating the Eq. (13). It should be clarified that **we only load the checkpoint at the beginning of the current dataset training process**.

---

> ### Author Response · Authors · 2024-11-22
>
> For **Question 5** about ''The framework caption in Figure 3 could benefit from more detailed explanations'':
>
> Thanks for your valuable suggestion, we have added more detailed explanations for Figure 3 and further improved the comprehensibility of the image.

---

> ### Author Response · Authors · 2024-11-22
>
> For **Question 6** about ''There are inconsistencies in the implementation details'':
>
> Thanks for your advice. There is indeed a bit of confusion, we have made a note of it. All experiments except Table 3 are conducted on 8-A100. Considering that Table 3 does not involve specific performance metrics (*e.g.* ACC/Forgetting), and the 8-A100 machine was occupied during the production of Table 3, thus we tested the training time and training parameters on the 8-A6000.

---

> ### Author Response · Authors · 2024-11-22
>
> For **Question 7** about ''The term "baseline" is used throughout the paper'':
>
> 1. Thanks for your advice. The term "baseline" consistently refers to LoRA fine-tuning, which has been clarified in Line 357-359. In Table 1, we also illustrated that the baseline equals the LoRA fine-tuning, masked as LoRA (Baseline) in the “Method” column. Besides that, we have added the further explanation that in the whole paper, baseline refers to the LoRA fine-tuning.
>
> 2. We compare the zero-shot performance (Table 4) of the stronger PGP method (in your suggestion) with ours under instruction type 1. Here are the results. The first column represents the continual training tasks. The second column represents the average accuracy of zero-shot inference datasets with PGP. The third column represents the average accuracy of zero-shot inference datasets with ours (here, we have the same zero-shot inference setting as Table 4).
>
> | Training Task | Zero-shot Performance of PGP | Zero-shot Performance of Ours |
> | :-----------: | :--------------------------: | :---------------------------: |
> | **ScienceQA** |            45.39             |           **47.55**           |
> |  **TextVQA**  |            39.96             |           **45.45**           |
> |    **GQA**    |            42.15             |           **45.24**           |
> |  **VizWiz**   |            38.78             |           **43.62**           |
> |   **VQAv2**   |            26.09             |           **28.58**           |
> |  **OCRVQA**   |             7.18             |           **7.82**            |
>
> It can be seen that our method owns better zero-shot performance than PGP.

---

> ### Author Response · Authors · 2024-11-22
>
> For **Question 8** about ''The term "but results for unseen tasks are missing'':
>
> Thanks for your valuable suggestion. For the convenience of comparison, we have placed the unseen (zero-shot) task performance (**marked with bold**) and seen task performance in one table (as shown below). The horizontally arranged dataset represents the test dataset, while the vertically arranged dataset represents the training dataset.
>
> For example, the value in the third row and the fourth column means the accuracy of the model trained on TextVQA and tested on VizWiz. Another example is that the value in the fourth row and the fifth column means the accuracy of the model trained on GQA and tested on VQAv2. Notice that baseline equals the LoRA Fine-Tuning.
>
> LLaCA Instruction Type 1
>
> |               |  TextVQA  |    GQA    |  VizWiz   |   VQAv2   |  OCRVQA   | ImageNet  |
> | :-----------: | :-------: | :-------: | :-------: | :-------: | :-------: | :-------: |
> | **ScienceQA** | **52.33** | **55.82** | **46.40** | **60.85** | **47.66** | **22.24** |
> |  **TextVQA**  |   59.69   | **53.52** | **43.00** | **60.28** | **52.43** | **18.02** |
> |    **GQA**    |   57.96   |   61.58   | **46.54** | **63.46** | **59.59** | **11.35** |
> |  **VizWiz**   |   58.34   |   60.52   |   52.00   | **65.91** | **56.55** | **8.40**  |
> |   **VQAv2**   |   58.10   |   60.00   |   36.47   |   66.78   | **48.64** | **8.51**  |
> |  **OCRVQA**   |   56.54   |   60.18   |   47.16   |   65.83   |   64.45   | **7.82**  |
>
> LLaCA Instruction Type 2
>
> |               |  TextVQA  |    GQA    |  VizWiz   |   VQAv2   |  OCRVQA   | ImageNet  |
> | :-----------: | :-------: | :-------: | :-------: | :-------: | :-------: | :-------: |
> | **ScienceQA** | **52.81** | **56.26** | **37.30** | **61.02** | **27.22** | **21.47** |
> |  **TextVQA**  |   59.37   | **59.17** | **34.29** | **60.29** | **51.60** | **17.90** |
> |    **GQA**    |   57.88   |   61.85   | **44.48** | **63.46** | **59.75** | **10.55** |
> |  **VizWiz**   |   58.51   |   60.46   |   50.29   | **65.90** | **57.72** | **7.84**  |
> |   **VQAv2**    |   58.56   |   59.83   |   36.68   |   66.57   | **52.14** | **8.22**  |
> |  **OCRVQA**   |   57.15   |   60.38   |   40.19   |   66.48   |   65.27   | **7.43**  |
>
> LoRA Instruction Type 1
>
> |               | TextVQA  |   GQA    |  VizWiz   |   VQAv2   |  OCRVQA   | ImageNet |
> | :-----------: | :------: | :------: | :-------: | :-------: | :-------: | :------: |
> | **ScienceQA** | **3.16** | **0.05** | **32.75** | **0.07**  | **0.00**  | **1.02** |
> |  **TextVQA**  |  50.63   | **4.37** | **18.85** | **2.99**  | **0.01**  | **0.67** |
> |    **GQA**    |  25.96   |  56.56   | **33.71** | **42.93** | **19.13** | **5.49** |
> |  **VizWiz**   |  26.96   |  39.31   |   55.68   | **40.03** | **10.05** | **3.58** |
> |   **VQAv2**    |  38.42   |  49.19   |   25.72   |   63.47   | **31.74** | **2.31** |
> |  **OCRVQA**   |  32.38   |  38.62   |   29.27   |   45.11   |   50.96   | **3.75** |
>
> LoRA Instruction Type 2
>
> |               | TextVQA  |   GQA    |  VizWiz   |   VQAv2   |  OCRVQA   | ImageNet |
> | :-----------: | :------: | :------: | :-------: | :-------: | :-------: | :------: |
> | **ScienceQA** | **3.92** | **0.37** | **42.49** | **0.95**  | **0.00**  | **1.79** |
> |  **TextVQA**  |  50.75   | **3.56** | **16.07** | **2.47**  | **2.85**  | **1.04** |
> |    **GQA**    |  30.16   |  58.02   | **22.23** | **44.43** | **21.09** | **6.35** |
> |  **VizWiz**   |  26.30   |  41.72   |   54.25   | **39.59** | **26.65** | **7.13** |
> |   **VQAv2**   |  39.87   |  49.28   |   22.64   |   63.77   | **33.12** | **5.20** |
> |  **OCRVQA**   |  32.34   |  38.02   |   15.33   |   44.42   |   56.18   | **4.34** |
>
> Zero-shot performance of original LLaVA, please kindly refer to **Table 1** and **Table 2** in the paper.
>
> **Our model shows a significant improvement in zero-shot performance compared to LoRA-Tuning and original LLaVA**, which we believe is due to the following reasons:
>
> (1). In terms of anti-forgetting during continual instruction tuning, considering that **our model can effectively protect old knowledge** (including pre-trained knowledge), our model owns better zero-shot capability compared to LoRA-Tuning.
>
> (2). In terms of learning how to generalize existing knowledge to new instructions, considering that **the instruction templates in each fine-tuning task are similar**, such as:
>
> TextVQA: Use a single word or a short phrase to respond to the question.
>
> VQAv2: Answer the question with a single word or a brief phrase.
>
> Therefore, compared to the original LLaVA, our model owns better zero-shot capability after fine-tuning.

---

> ### Author Response · Authors · 2024-11-22
>
> For **Question 9** about ''The paper cites many arXiv preprints'':
>
> Thanks for your valuable suggestion, we have rectified the reference form which has been published.

---

> ### Author Response · Authors · 2024-11-22
>
> For **Question 10** about ''Using a single equation number for each derivation would be clearer'':
>
> Thanks for your valuable suggestion, we have utilized a single equation number for each derivation.

---

### Official Review · Reviewer_ubn8 · 2024-11-01

**Soundness:** 3
**Presentation:** 3
**Contribution:** 3
**Rating:** 6
**Confidence:** 4

**Summary:**

Instruction tuning is currently a very hot topic, as it guides the Multimodal Large Language Models (MLLMs) in aligning different modalities by designing text instructions. In this framework, Multimodal Continual Instruction Tuning (MCIT) is adopted to continually instruct MLLMs to follow human intent in sequential datasets.

**Strengths:**

1.	Authors introduce a novel framework LLaCA for Multimodal Continual Instruction Tuning (MCIT) based on Exponential Moving Average (EMA) update policy.
2.	LLaCA overcomes the disadvantage of previous MCIT methods that require a great amount of trainable parameters, achieving parameter-efficiency during training.
3.	LLaCA exhibits good performance on various QA datasets and lowest forgetting score with both two different instruction types.

**Weaknesses:**

1.	While LLaCA outperforms several baselines, LLaCA is only evaluated on QA tasks (ScienceQA, TextVQA, GQA, VizWiz, VQAv2, OCRVQA), covering perception, recognition, OCR, and visual reasoning capabilities of MLLMs. More tasks such as Visual Entailment (VE) and classic classification (CIFAR100, MNIST) could be considered as a part of performance evaluation to further validate the effectiveness of LLaCA.

2.	LLaCA performs well with LoRA, which is remarkable. However, the performance of LLaCA with different PEFT methods such as Prompt-tuning [ref1] and Adapter could also be explored and included. The inherent differences between the different PEFT methods need to be highlighted and further discussed.

3.	All experiments conducted in the paper are on a single LLaVA-1.5 architecture. Suggestion: more backbone models, including Mini-GPT4 [ref2], OpenFlamingo [ref3] could be employed for more comprehensive studies of LLaCA.

4.	Discussion on Optimization Analysis such as Attention Activation Pattern Analysis [ref4] and Loss Landscape [ref5] could be beneficial for investigating why LLaCA holds superior performance compared to the other baselines.

[ref1] M^2PT: Multimodal Prompt Tuning for Zero-shot Instruction Learning

[ref2] MiniGPT-4: Enhancing Vision-Language Understanding with Advanced Large Language Models

[ref3] OpenFlamingo: An Open-Source Framework for Training Large Autoregressive Vision-Language Models

[ref4] Massive Activations in Large Language Models

[ref5] Visualizing the Loss Landscape of Neural Nets

**Questions:**

Please see the Weaknesses.

---

> ### Author Response · Authors · 2024-11-22
>
> Dear reviewer ubn8, thanks for your valuable suggestions. Here are our responses:
>
> For **Weakness 1** about ''further validate the effectiveness of LLaCA'':
>
> Thanks for your valuable suggestion, we add one image classification task (ImageNet-1K) and one Grounding task (based on the COCO dataset) in the MCIT task. The new tuning sequence order is as follows:
> **ScienceQA->TextVQA->ImageNet->GQA->VizWiz->Grounding ->VQAv2->OCRVQA**, with a total of 8 tasks. The following are the experimental results:
>
> Instruction Type 1
>
> | Method  | ScienceQA | TextVQA | ImageNet |  GQA  | VizWiz | Grounding | VQAv2 | OCRVQA | Avg. ACC  | Forgetting |
> | :-----: | :-------: | :-----: | :------: | :---: | :----: | :-------: | :---: | :----: | :-------: | :--------: |
> |  LoRA   |   49.28   |  31.19  |  10.36   | 36.54 | 28.03  |   1.42    | 46.35 | 52.87  |   32.01   |   27.38    |
> | MoELoRA |   63.09   |  38.63  |  10.50   | 37.38 | 43.62  |   0.59    | 43.15 | 60.08  |   37.13   |   25.91    |
> |  Ours   |   75.63   |  54.47  |  43.64   | 60.70 | 43.37  |   36.00   | 65.21 | 63.59  | **55.33** |  **7.04**  |
>
> Instruction Type 2
>
> | Method  | ScienceQA | TextVQA | ImageNet |  GQA  | VizWiz | Grounding | VQAv2 | OCRVQA | Avg. ACC  | Forgetting |
> | :-----: | :-------: | :-----: | :------: | :---: | :----: | :-------: | :---: | :----: | :-------: | :--------: |
> |  LoRA   |   42.17   |  30.25  |   9.68   | 36.41 | 26.22  |   0.56    | 42.93 | 58.19  |   30.80   |   32.76    |
> | MoELoRA |   47.34   |  32.91  |  10.12   | 37.15 | 42.48  |   0.97    | 42.77 | 57.50  |   33.91   |   28.55    |
> |  Ours   |   74.23   |  56.90  |  37.35   | 60.03 | 48.65  |   22.27   | 66.02 | 63.80  | **53.66** |  **9.62**  |
>
> It can be seen that under 8 instruction tuning tasks, our method still has better performances compared to Baseline (Inserted with LoRA) and MOELoRA [1]. It is worth noting that our judgment of whether the prediction results are correct or not is **strictly based on the direct comparison between outputs of MLLMs and ground truths**, which is defined as **Instruction Following Ability** in [1].  Therefore, our judgment criteria would be more stringent. It can be seen that under such strict standards, the baseline model (LLaVA+LoRA) experienced extremely severe catastrophic forgetting, but our model still has excellent performance.
>
> [1] COIN: A Benchmark of Continual Instruction Tuning for Multimodel Large Language Model

---

> ### Author Response · Authors · 2024-11-22
>
> For **Weakness 2** about ''The inherent differences between the different PEFT methods'':
>
> Thanks for your advice. We transfer our method from the LoRA Tuning paradigm to the **Adapter Tuning paradigm [2]**, leaving other settings unchanged.
>
> Instruction Type 1 (With Adapter Tuning)
>
> | ScienceQA | TextVQA | GQA   | VizWiz | VQAv2 | OCRVQA |
> | --------- | ------- | ----- | ------ | ----- | ------ |
> | 79.35     | 58.66   | 61.54 | 49.36  | 66.42 | 61.51  |
> | 74.54     | 55.91   | 61.42 | 45.71  | 66.79 | 61.51  |
>
> Avg.ACC = 60.98, Forgetting = 2.27, New ACC = 62.81
>
> Instruction Type 2 (With Adapter Tuning)
>
> | ScienceQA | TextVQA | GQA   | VizWiz | VQAv2 | OCRVQA |
> | --------- | ------- | ----- | ------ | ----- | ------ |
> | 78.69     | 57.74   | 61.75 | 50.15  | 66.28 | 60.08  |
> | 74.25     | 54.12   | 61.08 | 46.58  | 65.67 | 60.08  |
>
> Avg.ACC = 60.30, Forgetting = 2.58, New ACC = 62.45
>
> Overall, when compared with the performances between LoRA and Adapter, our method achieves **equivalent Avg.ACC performance**. However, there are **some differences in the Forgetting performance**. The reasons we analyze are as follows:
>
> (1). [3] deems that although Adapter and LoRA belong to two completely different fine-tuning paradigms, **their mathematical forms remain consistent**. Therefore, they should have **similar comprehensive performance (Avg.ACC) on the same anti-forgetting method**.
>
> (2). Specifically, the Adapter process inputs by adding a parallel bypass module, and finally merges the obtained results with the output of the original module, without any interaction with the original model parameters in this process. However, LoRA constructs a new matrix with the same dimensions as the original module by multiplying two low-rank matrices, and merges the new matrix with the original module parameters based on element-wise sum. It interacts with the original model parameters during this process. The essential mechanism of catastrophic forgetting is that the changing of model parameters disrupts the original parameter distribution. Therefore, **LoRA which generates interaction performs worse in anti-forgetting than Adapter which does not generate interaction**.
>
> [2] Introducing language guidance in prompt-based continual learning
>
> [3] Gradient Projection For Continual Parameter-Efficient Tuning

---

> ### Author Response · Authors · 2024-11-22
>
> For **Weakness 3** about ''more backbone models'':
>
> Thanks for your suggestion. Considering that both Mini-GPT4 [4] and OpenFlamingo [5] have significant differences in the input forms of data compared to LLaVA codebase, it is not possible to deploy our method on these two frameworks in time. We will gradually implement this in the future. But, here, we choose InternVL [6] as another MLLM and deploy our method and continual instruction setting on it. The reason why we choose InternVL is that it has a more powerful visual encoder (6B InternViT) and a three-stage model pre-training phrase, which provides it with powerful multimodal information processing capabilities. **It is crucial to verify whether our method is effective on stronger models**. Here, we still choose to insert LoRA on the LLM side (7B) of InternVL in each layer as the baseline. Meanwhile, we add our dynamic EMA weight update method to the baseline as our method for comparison. The following table shows the experimental results.
>
> |   Metric   | LoRA-Instruction1 | Ours-Instruction1 | LoRA-Instruction2 | Ours-Instruction2 |
> | :--------: | :---------------: | :---------------: | :---------------: | :---------------: |
> |  Avg. ACC  |       46.74       |     **64.14**     |       43.51       |     **62.94**     |
> | Forgetting |       18.32       |     **2.04**      |       22.10       |     **2.49**      |
> |  New ACC   |       64.43       |     **65.84**     |       64.08       |     **65.01**     |
>
> It can be seen that **our method is also effective for InternVL, significantly improving its continual instruction tuning ability, which proves the generalization and effectiveness of our method for more MLLMs**.
>
> [4] MiniGPT-4: Enhancing Vision-Language Understanding with Advanced Large Language Models
>
> [5] OpenFlamingo: An Open-Source Framework for Training Large Autoregressive Vision-Language Models
>
> [6] InternVL: Scaling up Vision Foundation Models and Aligning for Generic Visual-Linguistic Tasks

---

> ### Author Response · Authors · 2024-11-22
>
> For **Weakness 4** about ''Discussion on Optimization Analysis'':
>
> Thank you for your valuable suggestion. It is very meaningful, and innovative, which indeed provides insight points. We haven't done any experiments related to Optimization Analysis before, and this is our first attempt to try it.
>
> Based on Landscape Analysis [7], we study the loss landscape maps of LLaCA and LoRA Fine-tuning on Task 1 (ScienceQA) and Task 2 (TextVQA). Our conclusion is that **in our method, the model can be continuously updated (Loss reduced to the minimum value area of Task 2) while ensuring that the Loss on Task 1 is near the minimum value area**.
>
> *For Plasticity*: we find that the loss landscape map of LLaCA is more convex than LoRA Fine-tuning both in Task 2 (TextVQA). It can be explained that LLaCA promotes flat minimizers and prevents the transition to chaotic behavior [7]. The corresponding phenomenon is that the New Task Accuracy of LLaCA on Task 2 is 59.69. However, the New Task Accuracy of LoRA Fine-tuning on Task 2 is 50.63. It can be seen that **due to the better convex of loss landscape map, LLaCA has better New ACC performance than LoRA Fine-tuning**.
>
> *For Stability*: In LoRA Fine-Tuning, it has been found that during the training of Task 1, the loss decrease to the minimum value region of the current task. When training Task 2, the loss also decreases to the minimum value area of the current task. However, on Task 1, the loss of model at this training task is in a larger value area. It means that the training of Task 2 disrupts the loss value of Task 1, causing it to move from a minimum value area to a larger value area, and resulting in poor performance of Task 2 trained model on Task 1 (catastrophic forgetting). However, in the LLaCA method, during the training of Task 1, the loss decreases to the minimum value region of the current task. When training Task 2, **constantly reviewing the model parameters on Task 1 is like a rope pulling the model parameters, so that the range of loss variation on Task 1 does not exceed the minimum value area too far**.
>
> [7] Visualizing the Loss Landscape of Neural Nets

---

> > ### Comment · Reviewer_ubn8 · 2024-11-26
> > **Thank you!**
> >
> > Thank you for the rebuttal. I read the responses from other reviews. I think this paper deserves a score of 6, so I maintain my score.

---

> > > ### Author Response · Authors · 2024-11-28
> > > **Thanks for Reviewer ubn8**
> > >
> > > Thank you for your diligent review of our paper. With your valuable comments and suggestions, especially for the Questions about extension experiments, we have significantly enlarged the applications of our paper. We sincerely appreciate you and your advice.

---

### Official Review · Reviewer_huvJ · 2024-11-03

**Soundness:** 2
**Presentation:** 2
**Contribution:** 2
**Rating:** 5
**Confidence:** 4

**Summary:**

The paper addresses the challenge of catastrophic forgetting in MCIT. While existing methods like EMA help keep prior knowledge, their fixed weights cannot adapt to evolving datasets. To address this, the authors introduce a novel framework, LLaCA. By deriving an optimal balance weight through Taylor expansion and the Lagrange multiplier method, LLaCA dynamically adjusts EMA weights, reducing memory and computation costs. Experimental results demonstrate that LLaCA achieves significant enhancements compared to baseline methods, highlighting its effectiveness in continual instruction tuning.

**Strengths:**

1. The paper is generally well-written and structured, followed by the motivation and explanation.
2. The model can be easily applied across a wide range of fine-tuning methods.
3. The model achieves good results compared to baseline methods, with only a modest increase in training time.

**Weaknesses:**

1. In the second equation of Eq(3), $\theta_{t}^*$ represents the parameters after training on the first $t-1$ datasets, and
$\theta_{t}^*$ represents the parameters after training on t datasets. When tuning on a new dataset, parameter updates are necessary; however, preserving performance on previous datasets does not require keeping parameters unchanged. I think it's better to maintain consistent loss values rather than preserving the parameters themselves.
2. The paper introduces the L1-Norm to approximate $\beta_t$. However, in line 291, the authors state that the L1-norm demonstrates exceptional performance without providing an explanation or presenting empirical results to support this claim.
3. While the paper states that it addresses continual learning in MLLMs, the proposed method does not leverage any modality-specific capabilities, such as distinct features for the feature encoder and the LLM. There's a question of whether $\beta_t$ should be calculated to balance across different modalities.
4. This method is applicable to LLMs without any modifications, and catastrophic forgetting in LLMs has been a well-studied area. The problem is not novel. Recent methods like [1] and [2] also address reducing catastrophic forgetting.
5. The paper does not compare its approach with SOTA methods for mitigating catastrophic forgetting in MLLMs, such as [3]. Most of the baselines used are designed for LLMs.

[1] Mitigating Catastrophic Forgetting in Large Language Models with Self-Synthesized Rehearsal \
[2] CURLoRA: Stable LLM Continual Fine-Tuning and Catastrophic Forgetting Mitigation \
[3] Model Tailor: Mitigating Catastrophic Forgetting in Multi-modal Large Language Models

**Questions:**

1. The paper presents a method for dynamically updating the EMA weight. However, it only compares results using a stable weight of 0.99 to show the method's effectiveness. I suggest adding additional results, such as those using other static weights or weight decay, to provide a more comprehensive evaluation.
2. The writing could be polished for clarity. For example, some detailed explanations could be moved out of the main text into the Appendix, such as the content on line 166 and Equation (4). Additionally, Section 4.4 is somewhat redundant and could be streamlined.

**Details Of Ethics Concerns:**

No ethics concerns

---

> ### Author Response · Authors · 2024-11-22
>
> Dear reviewer huvJ, thanks for your valuable suggestions. Here are our responses:
>
> For **Weakness 1** about ''maintain consistent loss values rather than preserving the parameters themselves'':
>
> Thanks for your suggestion. Here is a brief reason why we don’t maintain consistent loss values.
>
> (1) LLaCA is a **non-rehearsal** method (considering some scenarios, such as online learning or privacy protection, where we cannot obtain historical data. Additionally, saving historical data also faces severe storage space costs), therefore we are unable to calculate the loss of the current model on historical tasks, which we think is a more practical setting.
>
> (2) The second equation in Eq.(3) is only **an ideal condition** where unchanged parameters can best protect the model performance on historical tasks. In fact, model updates are necessary in our method. While we hope to minimize the impact of model updates on model performance on historical tasks through our method, thereby avoiding the influence on model stability. To this end, we introduce the first equation in Eq.(3), which can ensure the plasticity of the model. In conclusion, the two equations in Eq.(3) are actually a multi-task learning objective, enabling our model to have both good stability and plasticity simultaneously.

---

> ### Author Response · Authors · 2024-11-22
>
> For **Weakness 2** about ''L1-norm demonstrates exceptional performance'':
>
> Considering that the L2 regularization is the most widely used method, we replace the L1 regularization in our method with L2 regularization and conduct multimodal continual instruction tuning experiments under two different instructions. The results are shown as:
>
> |   Metric   | L1-Instruction1 | L2-Instruction1 | L1-Instruction2 | L2-Instruction2 |
> | :--------: | :-------------: | :-------------: | :-------------: | :-------------: |
> |  Avg. ACC  |    **61.89**    |      59.79      |    **61.23**    |      57.87      |
> | Forgetting |    **2.68**     |      4.70       |    **3.38**     |      7.16       |
> |  New ACC   |    **64.12**    |      63.70      |    **64.05**    |      63.84      |
>
> It can be seen that there is a significant difference in performance between L2 regularization and L1 regularization, and the L1 regularization results we chose have better performance than L2 regularization. Analyzing the reasons, we believe that it is because L1 penalizes the absolute values of the parameters, and the sign changes of the parameters have no effect on the regularization term. Compared to other regularization methods such as L2, **L1 regularization is more robust to outliers [1]** because it does not excessively penalize large weights, thus **maintaining the stability of EMA self-adaption weights**.
>
> [1] Robust L1 Norm Factorization in the Presence of Outliers and Missing Data by Alternative Convex Programming

---

> ### Author Response · Authors · 2024-11-22
>
> For **Weakness 3** about ''whether $\beta_t$ should be calculated to balance across different modalities'':
>
>
> Thanks for your valuable suggestion. To analyze the $\beta_t$ implicitly leverages the modality-specific information, we conduct the following ablation experiments (conducted under Instruction Type 1). 1): **Normal**. Use the normal gradient update method to update the projection layer from the visual model to the language model and LoRA in LLM. 2): **Dy(Proj)**. Use dynamic EMA update (our method) to update the projection layer from the visual model to the language model, while updating the LoRA in LLM with the normal gradient update method. 3): **Dy(LLM)**. Use dynamic EMA update (our method) to update the LoRA in LLM, while updating the projection layer from the visual model to the language model with the normal gradient update method. 4): **Dy(Proj+LLM)**. Use dynamic EMA update (our method) to update the LoRA in LLM and the projection layer from the visual model to the language model. 5): **Dy(Vis)**. Extra insert LoRA into the vision encoder and use dynamic EMA update (our method) to update the LoRA in the vision encoder, while updating the projection layer from the visual model to the language model, and LoRA in LLM with the normal gradient update method. 6): **Dy(Proj+LLM+Vis)**. Extra insert LoRA into the vision encoder and use dynamic EMA update (our method) to update the LoRA in the vision encoder, LoRA in LLM, and the projection layer from the visual model to the language model. The results are shown as:
>
> |   Metric   | Normal | Dy(Proj) | Dy(LLM) | Dy(Proj+LLM) | Dy(Vis) | Dy(Proj+LLM+Vis) |
> | :--------: | :----: | :------: | :-----: | :----------: | :-----: | :-----------------: |
> |  Avg. ACC  | 41.31  |  55.87   |  61.83  |    61.89     |  50.72  |        62.38        |
> | Forgetting | 22.67  |  11.12   |  2.96   |     2.68     |  13.24  |        2.31         |
> |  New ACC   | 60.20  |  65.13   |  64.30  |    64.12     |  63.85  |        64.31        |
>
> It can be seen that **adding our proposed method at any position in the LLaVA model can help further improve the performance of MCIT**. The next is the key observations: :
>
> (1) When training the projection layer, it is equivalent to **mapping the encoded visual features to the language-text space**. Both the first-order and second-order gradients of the projection layer collect the information from two modalities. Therefore, when calculating the EMA update weight of the projection layer, it can be seen as a result obtained on the basis of balancing the two modalities.
>
> (2) Similarly, after concatenating the encoded visual features and the textual features into union features, we input them into LLM. Therefore, **the features processed by LoRA are a mixture of visual and textual features**. it can also be seen as a result obtained on the basis of balancing the two modalities.
>
> (3) In addition, we have also tried using other methods to obtain modality-specific capabilities, such as directly sampling the feature space of the vision encoder and LLM, but the computational cost is expensive. Considering that we have inserted LoRA parameters in each layer of LLM, we need to sample the feature space at every layer, which is also an unbearable computational cost.
>
> (4) It is worth mentioning that although LoRA is also inserted into the vision encoder in **Dy(Proj+LLM+Vis)** and dynamically updated according to the proposed method, its performance is slightly better than that in **Dy(Proj+LLM)**. Similarly, in **Dy(Vis)**, we also insert additional LoRA into the vision encoder and update it according to the proposed method, while keeping other trainable parameters updated with the normal gradient update method. It is still inferior to the results in **Dy(Proj)** and **Dy(LLM)**. The above experiments all explain that **in the case of a single modality (image), our method has little effect and can achieve maximum benefits in multimodal scenarios**. This also reflects indirectly that $\beta_t$ is calculated to balance across different modalities.

---

> ### Author Response · Authors · 2024-11-22
>
> For **Weakness 4** about ''This method is applicable to LLMs without any modifications'':
>
> 1. Please kindly note that [2] actually synthesizes the historical text data based on LLMs for rehearsing, but this process **requires additional training time and computing power (synthesizing pseudo rehearsal data)**. In addition, the method in [2] is not applicable for the cross-modality continual instruction tuning task, as the image encoder of the multimodal large language modal **can only be used to compress features and cannot be used to generate/synthesize images**. On the other hand, this is a very different solution for anti-forgetting, which implies [2] could cooperate with our approach in some data-synthesized scenes.
>
> 2. By comparison, our method belongs to the non-rehearsal method, which has the advantages of 1) avoiding the need to save or generate rehearsal samples, reducing computational costs and memory space; 2) No need to train additional rehearsal examples, shortening the overall training time. 3) Strong generality, not limited by any tasks, scenarios, or models.
>
> 3. To further compare with [3], we transfer our method to the continual fine-tuning framework in [3] (Based on Mistral 7B (v0.3) [4]) and conduct continual tuning experiments on the Continual Classification Benchmark (MRCC->SST-2->Sentiment140). The following is the comparison of the experimental results (CURLORA results are reported from the original paper). **It can be seen that our method has equal stability and better plasticity in LLM continual fine-tuning tasks compared to the method in [3]**. The first column in the two tables represents the accurate values obtained at the current task, while the second column represents the accurate values obtained at the final task.
>
> Ours
> | **MRPC** | **SST-2** | **Sentiment140** |
> | :------: | :-------: | :--------------: |
> |  0.6716  |  0.9025   |       1.00       |
> |  0.7353  |  0.9025   |       1.00       |
>
> CURLORA
> | **MRPC** | **SST-2** | **Sentiment140** |
> | :------: | :-------: | :--------------: |
> |   0.66   |   0.86    |       0.94       |
> |   0.66   |   0.86    |       0.94       |
>
> 4. Finally, we transfer the method in [3] to the MCIT framework used in our paper and conduct continual instruction tuning experiments on six VQA datasets (under Instruction Type 1). The following is a comparison of the experimental results.
>
> | Method  | ScienceQA | TextVQA |  GQA  | VizWiz | VQAv2 | OCRVQA |    Avg. ACC    |   Forgetting    |    New ACC    |
> | :-----: | :-------: | :-----: | :---: | :----: | :---: | :----: | :-------: | :------: | :-------: |
> |  Ours   |   77.15   |  56.54  | 60.18 | 47.16  | 65.83 | 64.45  | **61.89** | **2.68** | **64.12** |
> | CurLoRA |   71.26   |  57.59  | 61.20 | 43.69  | 66.51 | 65.23  |   60.91   |   3.71   |   64.00   |
>
> Experiments in 3 and 4 demonstrate that **our method outperforms than method in [3]**.
>
> 5. By the way, the method in [3] only focuses on the LoRA fine-tuning paradigm and cannot be well transferred to other fine-tuning paradigms such as Adapter [5]. By comparison, our method can seamlessly transfer to other fine-tuning paradigms (the experimental results of replacing LoRA Tuning with Adapter Tuning are shown in the table below). It can be seen that **our method has considerable performance under the LoRA and Adapter fine-tuning paradigms, and even more other Parameter Efficient Fine-Tuning (PEFT) Paradigms**.
>
> Instruction Type 1 (With Adapter Tuning)
>
> | ScienceQA | TextVQA | GQA   | VizWiz | VQAv2 | OCRVQA |
> | --------- | ------- | ----- | ------ | ----- | ------ |
> | 79.35     | 58.66   | 61.54 | 49.36  | 66.42 | 61.51  |
> | 74.54     | 55.91   | 61.42 | 45.71  | 66.79 | 61.51  |
>
> Avg.ACC = 60.98, Forgetting = 2.27, New ACC = 62.81
>
> Instruction Type 2 (With Adapter Tuning)
>
> | ScienceQA | TextVQA | GQA   | VizWiz | VQAv2 | OCRVQA |
> | --------- | ------- | ----- | ------ | ----- | ------ |
> | 78.69     | 57.74   | 61.75 | 50.15  | 66.28 | 60.08  |
> | 74.25     | 54.12   | 61.08 | 46.58  | 65.67 | 60.08  |
>
> Avg.ACC = 60.30, Forgetting = 2.58, New ACC = 62.45
>
> [2] Mitigating Catastrophic Forgetting in Large Language Models with Self-Synthesized Rehearsal
>
> [3] CURLoRA: Stable LLM Continual Fine-Tuning and Catastrophic Forgetting Mitigation
>
> [4] Mistral 7b
>
> [5] Introducing language guidance in prompt-based continual learning

---

> ### Author Response · Authors · 2024-11-22
>
> For **Weakness 5** about "The paper does not compare its approach with SOTA methods for mitigating catastrophic forgetting in MLLMs'':
>
> We transfer the method in [6] to the MLLM continual tuning framework used in this paper, and the experimental results are compared as follows:
>
> |   Metric   | Ours-Instruction1 | MT-Instruction1 | Ours-Instruction2 | MT-Instruction2 |
> | :--------: | :---------------: | :-------------: | :---------------: | :-------------: |
> |  Avg. ACC  |     **61.89**     |      56.88      |     **61.23**     |      54.35      |
> | Forgetting |     **2.68**      |      11.28      |     **3.38**      |      13.39      |
> |  New ACC   |     **64.12**     |      66.28      |     **64.05**     |      65.51      |
>
> It can be seen that our method outperforms the method in [6] with two instruction templates and three metrics (Avg. ACC, Forgetting, and New. ACC). We hold the view that the method proposed in [6] is only suitable for one or two downstream fine-tuning tasks, as its **starting point is to protect pre-trained knowledge and alleviate its forgetting during downstream task generalization**. Therefore, it is not suitable for multiple continual tuning tasks (such as six tasks in our experimental setting).
>
> As emphasized in **Appendix A. More Details about Related Work [6]**, the author claims that “*while (He et al., 2023) focuses on sequential tasks and measuring the forgetting of older tasks when learning new tasks, our focus is on the forgetting of the pre-trained MLLM after fine-tuning specific tasks*”. Different from their goals, we focus on preserving knowledge of historical tasks (**including both pre-training tasks and continual fine-tuning tasks**) and mitigating catastrophic forgetting during the fine-tuning process.
>
> It is worth noting that our method can also be directly applied to the setting in [6], which can be seen as a sub-task of multimodal continual instruction tuning (one-step continual learning). A similar task can be seen in **Table 4** in our paper (The difference is that [6] test on the pre-training tasks, while we test on other unseen tasks). Besides that, considering the time constraints, we will try the setting in [6] with our method in future work.
>
> [6] Model Tailor: Mitigating Catastrophic Forgetting in Multi-modal Large Language Models

---

> ### Author Response · Authors · 2024-11-22
>
> For **Question1** about ''more comprehensive evaluation'':
>
> Thanks for your advice. Based on Instruction Type 1, we measure **four fixed EMA weights** for comparison, *e.g.* 0.500 0.990 0.991 0.992. In addition, we also measure the experimental results of **a naive EMA weight decay method** ($ \beta_t=0.9991+0.0002 (t-2), t > 1$, while $t=1$ without EMA update). Finally, to further supplement our conclusion, we also measure the experimental results based on the **parameter weight decay (EWC) method**. To sum up, our method is still superior to the above methods.
>
> |   Metric   | 0.500 | 0.990 | 0.991 | 0.992 | EMA Decay |  EWC  |   Ours    |
> | :--------: | :---: | :---: | :---: | :---: | :-------: | :---: | :-------: |
> |  Avg. ACC  | 58.96 | 60.76 | 58.83 | 60.69 |   59.07   | 44.49 | **61.89** |
> | Forgetting | 8.68  | 5.17  | 7.06  | 5.46  |   5.94    | 12.69 | **2.68**  |
> |  New ACC   | 66.20 | 65.08 | 64.71 | 65.24 |   64.02   | 55.07 | **64.12** |

---

> ### Author Response · Authors · 2024-11-22
>
> For **Question2** about ''The writing could be polished for clarity'':
>
> Thanks for your valuable suggestion, we have moved some detailed explanations from the main text to the Appendix. Additionally, we also simplify the Section 4.4.

---

> > ### Comment · Reviewer_huvJ · 2024-11-27
> >
> > Thank you for the rebuttal.
> >
> > I am still a little bit confused about the second equation in Eq(3). I know it's the ideal case, but updating parameters is required during training. Are you assuming that different parameters are only applied to specific datasets? For example, when training on the first dataset, there are parameters specific to dataset one, and then for the second dataset, new parameters are introduced for that dataset? If so, I don’t think this approach is realistic in real-world scenarios.
> >
> > For weakness 3, the authors mention that "adding our proposed method at any position in the LLaVA model can help further improve the performance of MCIT." Does this mean the proposed method can also be used to update LoRA in any LLM, not just in multimodal? Could you clarify your contribution specific to multimodal? Specifically, how does your method help balance different modalities during continual learning across different datasets? One example as, it involves projecting the vision encoder into a different space.
> >
> > If the author can address my concern, I would like to raise my rating.

---

> > > ### Author Response · Authors · 2024-11-28
> > > **Responses to The Questions**
> > >
> > > Dear reviewer huvJ, thanks for your reply.
> > >
> > > **For your first question:**
> > >
> > > No, we **don’t assume** that different parameters are only applied to specific datasets. In our method, we continually train the **same parameters** on different datasets, and the parameters are **task-shared**. Please kindly refer to Line 92-94 and Figure 2(a) in our paper. In the test phase, we use the **same parameters** to infer the different multimodal tasks. The whole training and test process are shown in Appendix A.11. Algorithm of LLaCA.
> > >
> > > **For your second question:**
> > >
> > > Thanks for your valuable suggestion. We think our method can be transferred to update LoRA in LLM with some small modifications.
> > >
> > > In fact, it is noticed that many existing training paradigms of MLLMs freeze the visual encoder and only train the parameters of LLM and projection layer to achieve good multimodal comprehension ability such as LLaVA and other SOTA models[1-3].
> > > Therefore, our method is based on this straight-forward training pipeline with our multimodal continual instruction tuning algorithm.
> > >
> > > However, we think this simple training pipeline may implicitly help balance different modalities. For a typical MLLMs, the image is embedded into visual features, and then projected by projection layer into language/text space. After that, the embedded text features are concatenated with projected visual features, and then decoded by LLM.
> > >
> > > 1. **Our method helps the projection layer progressively learn how to project distinct visual features to the language-text space on incremental tasks**. Across different datasets, we continually train one projection layer. In the Table we listed to answer Weakness 3, we can clearly find that, compared with normal gradient update (named as “Normal” in the Table), using our proposed method to update the projection layer (named as “Dy(Proj)” in the Table) can help greatly improve the performance. Thus, our method really helps balance different modalities during continual learning, when it involves projecting the encoded visual features to the language-text space.
> > >
> > > 2. **Our method helps the LLM progressively learn how to understand the diverse projected visual features on incremental tasks**. Across different datasets, we continually train one LoRA parameter inserted in the LLM. In the Table we listed to answer Weakness 3, we can clearly find that, compared with normal gradient update (named as “Normal” in the Table), using our proposed method to update the LoRA (named as “Dy(LLM)” in the Table) can help greatly improve the performance. Thus, our method really helps balance different modalities during continual learning, when it involves assisting the LLM to understand the projected visual features with inserted LoRA.
> > >
> > > In our knowledge, MLLM methods often focus on utilizing stronger and bigger vision or projection structure to explicitly balance different modalities. For example: 1. For the vision encoder, [3] replaces the hundreds of MB ViT with 6B InterViT, 2. For the projection layer, [4] replaces two linear layers with hundreds of MB Q-Former [4]. We sincerely thank the reviewer for providing valuable suggestions and insights, which provoke an in-depth examination of the contributions of our method in multimodal continual learning.
> > >
> > > [1]Visual Instruction Tuning
> > >
> > > [2]MiniGPT-v2: Large Language Model As a Unified Interface for Vision-Language Multi-task Learning
> > >
> > > [3]InternVL: Scaling up Vision Foundation Models and Aligning for Generic Visual-Linguistic Tasks
> > >
> > > [4]BLIP-2: Bootstrapping Language-Image Pre-training with Frozen Image Encoders and Large Language Models

---

> ### Comment · Reviewer_huvJ · 2024-12-03
>
> Thanks for your response to my question. While I still have concerns about the scope, I appreciate the authors' hard work during the rebuttal process and have raised my score to 5 accordingly.

---

> > ### Author Response · Authors · 2024-12-04
> > **Thanks for Reviewer huvJ**
> >
> > Thank you for your hard work in reviewing our paper. We sincerely appreciate your valuable comments and suggestions, such as applying our method to different application scenes, e.g. NLP, and the inspirations in multimodal continual learning, which enhance the quality of our paper.

---

### Meta-Review · Area_Chair_tSmv · 2024-12-22

**Metareview:**

This paper addresses the problem of catastrophic forgetting in MCIT for MLLMs by proposing LLaCA, a method that employs a dynamic Exponential Moving Average strategy. LLaCA aims to balance retaining prior knowledge with acquiring new information during continual learning while minimizing additional computational overhead.

Although the paper addresses an important issue, it lacks sufficient novelty in the multimodal domain. The approach primarily adapts a general continual learning technique without adequately tackling the unique challenges of multimodal learning, such as modality interaction and fusion. The experimental evaluation is limited, restricting validation of the method's effectiveness across diverse scenarios, and the paper lacks thorough comparisons with recent methods. Additionally, critical theoretical justifications require further explanation or empirical validation. Combined with presentation issues that affect readability and comprehension, these weaknesses suggest that the paper, in its current form, does not meet the threshold for acceptance.

**Additional Comments On Reviewer Discussion:**

During the reviewer discussion, several key concerns were raised about the paper’s contributions, evaluation scope, novelty, and presentation clarity.

Reviewers huvJ and RGJd noted that LLaCA primarily adapts general continual learning techniques without adequately addressing the unique challenges of multimodal learning, such as modality interaction and fusion. They questioned the method's novelty in the multimodal domain and highlighted the lack of theoretical justification, calling for stronger explanations or empirical validation. RGJd also emphasized the absence of comparisons with stronger methods and noted presentation issues affecting readability.

Reviewer ubn8 observed that the experimental evaluation focused mainly on visual question answering tasks and suggested expanding to additional tasks like image classification and grounding, as well as testing with alternative backbone models to improve generalizability. The authors’ expanded experiments addressed some of ubn8’s concerns, clarifying certain aspects of the evaluation.

Despite the authors’ efforts to address these points through additional experiments and revisions, the paper in its current form lacks sufficient novelty, robust validation, and comprehensive comparisons, and thus does not meet the criteria for acceptance.

---

### Decision · Program_Chairs · 2025-01-22

Reject